# Automatic Reward Shaping from Multi-Objective Human Heuristics

## Abstract

Designing effective reward functions remains a central challenge in reinforcement learning, especially in multi-objective environments. In this work, we propose **M**ulti-**O**bjective **R**eward **S**haping with **E**xploration, a general framework that automatically combines multiple human-designed heuristic rewards into a unified reward function. MORSE formulates the shaping process as a bi-level optimization problem: the inner loop trains a policy to maximize the current shaped reward, while the outer loop updates the reward function to optimize task performance. To encourage exploration in the reward space and avoid suboptimal local minima, MORSE introduces stochasticity into the shaping process, injecting noise guided by task performance and the prediction error of a fixed, randomly initialized neural network. Experimental results in MuJoCo and Isaac Sim environments show that MORSE effectively balances multiple objectives across various robotic tasks, achieving task performance comparable to those obtained with manually tuned reward functions.

## 1 Introduction

Reward design remains a fundamental challenge in robot learning, particularly when multiple conflicting objectives must be balanced. A typical task definition might involve the robot traversing a specified distance (e.g., 2 meters) within a given time limit (e.g., 10 seconds) while maintaining the torque below a safety limit (e.g., 45 Nm). While the success criteria for the task are straightforward to define, their sparsity makes them inadequate as rewards for effective policy optimization. (Andrychowicz et al., 2018)

To facilitate policy learning, researchers often design dense heuristic reward functions, such as rewarding higher velocities or penalizing large joint movements to minimize energy cost. However, these heuristics frequently conflict with one another, necessitating labor-intensive manual tuning to combine them into a final reward function. Studies show that over $90\%$ of reinforcement learning (RL) practitioners devote substantial effort to adjusting reward functions, highlighting a critical scalability bottleneck (Booth et al., 2023).

Our work proposes a self-supervised reward shaping framework that automates the reward design process. The approach requires RL practitioners to specify only: (1) a task performance criterion, which scores a trajectory at the end of the episode, and (2) a set of heuristic reward functions, which provide dense feedback for each step. The system then automatically learns an optimal combination of these heuristics through a bi-level optimization process (Zhang et al., 2024b), where the inner loop trains RL policies under the current reward function, and the outer loop adapts the reward function itself to maximize task performance.

However, it is not trivial to apply bi-level optimization to shape the rewards for real-world robotic tasks. Consider the robotic dog benchmark in (Margolis & Agrawal, 2023), which defines 15 distinct objectives, constituting an enormous space of valid reward combinations. With such a large reward weight space, the weight–performance landscape becomes highly non-convex with numerous local extrema (Xie et al., 2024). Under such circumstances, existing bi-level methods (Gupta et al., 2023) fail to converge reliably in complex scenarios, as we will further demonstrate in the experiments.

To overcome these challenges, we introduce **M**ulti-**O**bjective **R**eward **S**haping with **E**xploration (MORSE). MORSE augments the outer loop of bi-level optimization with controlled stochastic ex-

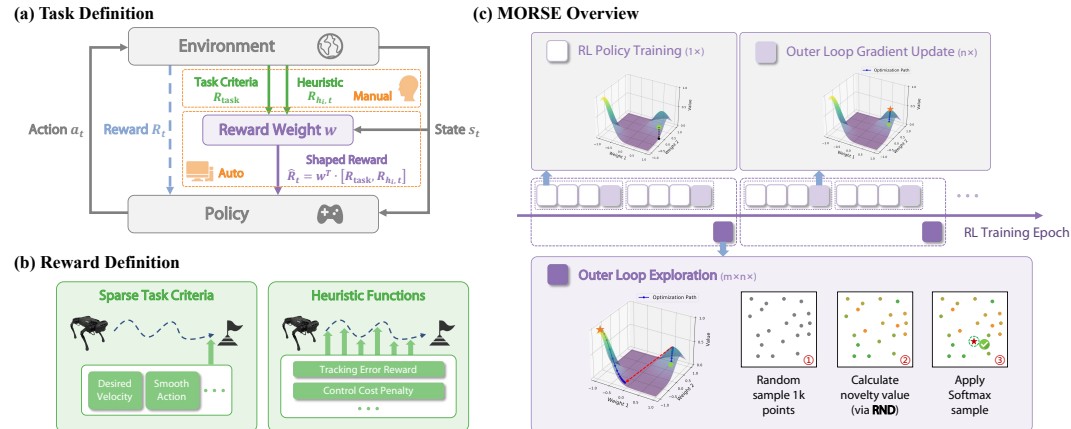

Figure 1: **(a)** Experts provide only task criteria and heuristic functions, instead of a manually tuned reward function. **(b)** Example task criteria and heuristic functions for robotic dog. **(c)** The inner loop trains the RL policy on shaped rewards, while the outer loop, combining gradient updates with exploration, updates reward weights to maximize task criteria. In the 3D graphs, we use xy plane to denote reward weight space, and z axis to represent the corresponding task performance.

ploration, thereby facilitating training when the reward space is large. Concretely, if the policy plateaus without reaching the target objective, the algorithm initiates outer-loop exploration: it samples a new reward weight and resumes gradient-based bi-level updates from this point. Exploration relies on novelty scores computed via Random Network Distillation (RND)(Burda et al., 2018), where the prediction error against a fixed, randomly initialized network serves as the novelty measure. New starting points are then selected through softmax sampling over these scores.

The experiments are organized as follows. We begin with a motivating example in a custom multi-objective CartPole environment and show that unguided exploration helps conventional bi-level approaches to escape from local optima. We then evaluate MORSE in two complementary settings. (1) Synthetic optimization surfaces, where we simplify bi-level optimization into single-level gradient update to isolate and validate each component of the controlled exploration mechanism. (2) Challenging robotics domains in MuJoCo (Towers et al., 2024) and Isaac Lab(Mittal et al., 2023), including a 9-objective quadruped locomotion task and a 5-objective robotic manipulation task. Across all environments, MORSE consistently outperforms vanilla bi-level optimization and achieves performance comparable to that obtained with human-engineered reward functions.

## 2 RELATED WORK

### 2.1 MULTI-OBJECTIVE REINFORCEMENT LEARNING

To incorporate the trade-offs between multiple objectives, different multi-objective reinforcement learning (MORL) methods have been proposed. These methods can be roughly divided into two categories, that is, single-policy and multi-policy (Hayes et al., 2022; Felten et al., 2023).

Single-policy methods yield a policy capable of adapting to any preference utility at test time. This can be done either during training, where the policy maximizes expected scalarized returns (ESR) for potential utilities, (Roijers et al., 2018; Siddique et al., 2020) or during execution, where the policy selects actions based on test-time utility(Van Moffaert et al., 2013a;b; Issabekov & Vamplew, 2012; Reymond et al., 2023).

Multi-policy methods, on the other hand, generate a collection of policies that form a Pareto Front or Coverage Set, capturing inter-objective relationships. They can iteratively sample utility functions and derive corresponding policies to construct an exhaustive coverage set (Parisi et al., 2014; Mossalam et al., 2016; Parisi et al., 2017; Natarajan & Tadepalli, 2005), or generate multiple policies concurrently and employ specific criteria such as hypervolume value or Monte Carlo Tree Search to select actions(Castelletti et al., 2012; Ruiz-Montiel et al., 2017; Van Moffaert & Nowé, 2014; Yang et al., 2019; Lu et al., 2022; Reymond et al., 2022).

In general, MORL focuses on recovering the Pareto front that captures the trade-offs between multiple objectives, but it does not learn how to choose or optimize the reward weights. It assumes that the appropriate weights are given at test time. Our method aims to solve the orthogonal problem. Our goal is to automatically adjust the reward weights of auxiliary rewards to better guide the RL policy toward higher success rates. Therefore, our problem setup fundamentally differs from MORL.

## 2.2 Reward Shaping in Reinforcement Learning

Designing effective reward functions poses fundamental challenges in RL: sparse rewards yield inefficient learning, while poorly shaped rewards may induce suboptimal behaviors. To address these challenges, existing reward shaping methods can be broadly categorized into four main approaches.

Heuristic-based intrinsic reward methods encourage exploration through visitation frequency counts (Tang et al., 2017; Lobel et al., 2023; Machado et al., 2020; Fu et al., 2017) or curiosity-driven uncertainty bonuses (Pathak et al., 2017; 2019; Zhang et al., 2020; Burda et al., 2018), but require careful manual tuning of intrinsic and extrinsic reward balances. More automated approaches include differentiable intrinsic reward learning (LIRPG Zheng et al. (2018)), trajectory preference-based reward inference (SORS Memarian et al. (2021) ), and composite reward functions combining task and exploration objectives (EXPLORS Devidze et al. (2022)). With the rapid development of large language models (LLMs), several recent works have explored using LLMs to assist in constructing reward functions(Ma et al., 2023; Zhang et al., 2024a). These approaches typically require the LLM to define both the heuristic functions and the reward weights, and to iteratively refine them over multiple optimization rounds, which leads to substantial computational overhead.

Most relevant to our work are bi-level optimization methods (Hu et al., 2020; Gupta et al., 2023) that formulate reward shaping as $\max_\phi \mathcal{R}_{task}(\pi_\theta^*)$ s.t. $\pi_\theta^* = \arg\max_\pi \mathcal{R}_\phi(s,a)$, where $\phi$ parameterizes the shaped reward. These method assume (i) smooth reward-performance landscapes and (ii) sufficient gradient signals from heuristic rewards. However, such assumptions are often violated in robotics tasks, causing outer-loop optimization to stagnate in local optima. Our method addresses these limitations by adding stochasticity that considers task performance and novelty measurement in reward weight space.

## 3 Preliminary

### 3.1 Task Formulation

We formulate each task as a Markov decision process (MDP) defined by a tuple $(\mathcal{S}, \mathcal{A}, \mathcal{P}, \hat{R}, \gamma)$, where $\mathcal{S}$ is the set of states, $\mathcal{A}$ is the set of actions, $\mathcal{P}$ is the state transition function, $\hat{R} \colon \mathcal{S} \times \mathcal{A} \to \mathbb{R}$ is the task-specific reward function, $\gamma$ is the discount factor. The goal is to find a policy $\pi_\theta$ that maximizes the discounted cumulative task reward, which is defined as $J(\theta) = \mathbb{E}_{\tau \sim \pi_\theta}[\Sigma_{t=0}^T \gamma^t \hat{R}(s_t, a_t)]$, where $\tau = (s_0, a_0, \ldots, s_T)$ denotes a trajectory sampled under $\pi_\theta$, and $T$ is the episode horizon.

Designing an effective reward function $\hat{R}$ is challenging, as it often requires balancing multiple conflicting objectives. In the proposed framework, we gradually learn $\hat{R}$ from (1) a sparse task-completion metric $R_{\text{task}}$ provided by a human expert (e.g., a binary indicator of success at episode termination) and (2) a set of $n$ auxiliary heuristic reward functions $\{R_{h_i}\}_{i=1}^n$, each capturing a separate objective of the task. Here, the human expert only needs to determine the terminal success criterion and provide heuristic guidance; the algorithm automatically finishes reward shaping.

Formally, the algorithm balances all reward components $\mathbf{R} = (R_{\text{task}}, R_{h_1}, \ldots, R_{h_n})$ to maximize task reward $R_{\text{task}}$. We learn a state-dependent weight function $w_\phi : \mathcal{S} \to \mathbb{R}^{n+1}$ to produce the composite reward:

$$R_\phi(s, a) = w_\phi(s)^\top \mathbf{R}(s, a),$$

where $\mathbf{R}(s, a) \in \mathbb{R}^{n+1}$ is the concatenated vector of all reward components.

## 3.2 BI-LEVEL OPTIMIZATION FOR REINFORCEMENT LEARNING

The above reward shaping process can be solved via bi-level optimization. The outer loop optimizes the reward weight $w_\phi(s)$ to maximize the accumulated task reward $J(\theta, \phi)$:

$$J(\theta, \phi) = E_{\pi_\theta, w_\phi}[\Sigma_{t=0}^T \gamma^t R_{\text{task}}(s_t, a_t)]$$

where as the inner loop optimizes the policy $\pi_\theta$ to maximize the weighted return $G(\theta)$:

$$G_\phi(\theta) = E_{\pi_\theta}[\Sigma_{t=0}^T \gamma^t R_\phi(s, a)] = E_{\pi_\theta}[\Sigma_{t=0}^T \gamma^t w_\phi(s_t) \cdot \mathbf{R}(s_t, a_t)]$$

The inner loop can be optimized using any reinforcement learning algorithm, while the outer loop can be solved using various bi-level optimization techniques. In this work, we choose the Implicit Function (IF)-based approach (Zhang et al., 2024b).

According to the chain rule, the gradient computation for the outer loop can be decomposed as:

$$\frac{\mathrm{d}J(\theta, \phi)}{\mathrm{d}\phi} = \frac{\partial J(\theta, \phi)}{\partial \phi} + \frac{\partial J(\theta, \phi)}{\partial \theta} \cdot \frac{\mathrm{d}\theta}{\mathrm{d}\phi}$$

Since $w_\phi$ does not explicitly influences $J(\theta, \phi)$, the term $\frac{\partial J(\theta, \phi)}{\partial \phi}$ equals zero. The next term $\frac{\partial J(\theta, \phi)}{\partial \theta}$ is the policy gradient calculated using the task reward, $R_{\text{task}}$. The third term $\frac{\mathrm{d}\theta}{\mathrm{d}\phi}$ captures how the optimal inner loop policy responds to changes in the reward weight $w_\phi$. It is determined by $\frac{\mathrm{d}\pi^*_{\theta;\phi}}{\mathrm{d}\phi}$, where $\pi^*_{\theta;\phi}$ is the optimal policy obtained in the inner loop with $w_\phi$.

We briefly discuss how to calculate the second term. During convergence, we have $\nabla_\theta G_\phi(\theta) = 0$, leading to the following expression for $\frac{\mathrm{d}\pi^*_{\theta;\phi}}{\mathrm{d}\phi}$:

$$\frac{\mathrm{d}\pi^*_{\theta;\phi}}{\mathrm{d}\phi} = -\left(\nabla_\phi^2 \mathbb{E}_\pi \left[\sum_{t=0}^T \gamma^t R_\theta\right]\right)^{-1} \nabla_\phi \mathbb{E}_\pi \left[\sum_{t=0}^T \gamma^t R_\theta\right]$$

The two terms above can be computed using the second-order and first-order derivatives of the inner loop objective function. However, computing the inverse Hessian matrix $H = \nabla_\phi^2 \mathbb{E}_\pi \left[\sum_{t=0}^T \gamma^t R_\theta\right]$ is computationally expensive. Therefore, we use the Neumann Series method to approximate the inverse of the Hessian matrix :

$$H^{-1} \approx \sum_{i=0}^{K-1} (I - H)^i$$

## 4 A MOTIVATION EXAMPLE: MULTI-OBJECTIVE CARTPOLE WITH EXTREMELY SPARSE REWARD

In this section, we extend the classic CartPole to a multi-objective setting where the agent must balance the pole while moving the cart as far right as possible. This demonstrates how vanilla bi-level optimization struggles with conflicting objectives and highly sparse task rewards. The environment retains the original observation and action spaces but uses a composite reward with four components: (1) a +100 task reward for surviving all 100 steps; (2) a +1 survival heuristic per step; (3) a +1 position heuristic when the cart's position exceeds 0.5; and (4) a -1 interference penalty when the agent's action matches a PD controller's stabilization attempt. These modifications make the multi-objective Cartpole significantly harder. We perform a grid search for potential reward weights, and the best performing configuration achieved approximately a 50% success rate.

We use REINFORCE in the inner loop for policy training, with the outer loop modeling rewards as $w(s) \cdot \mathbf{R}(s, a)$ through a weight network $w(s)$ that maps states to reward weights. We compare three update strategies for $w(s)$: **Constant** (randomly initialized weights with no update), **Gradient** (updated with standard bi-level optimization), and **Gradient w/ Reset** (resetting the weight network parameters every five gradient updates).

We evaluate each method over 10 trials with different random seeds, and report the results in Fig. 2a. Compared with **Constant**, **Gradient** improves the task success rate to 30%, while **Gradient w/ Reset**, which periodically resets the reward weight, further increases the success rate to 50%.

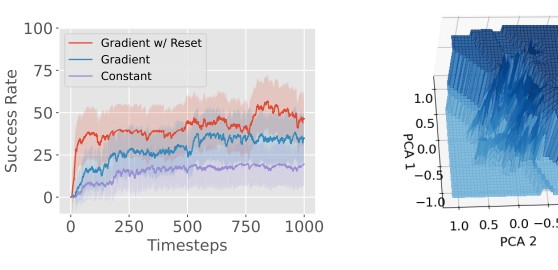 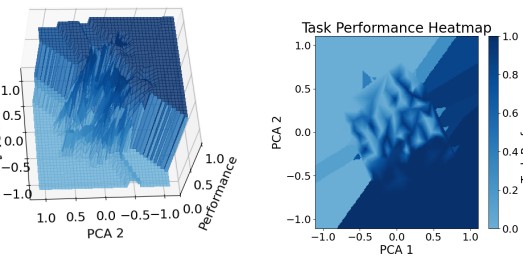

(a) **Gradient** failure case.                    (b) Weight–performance relation.

Figure 2: Lesson from the motivation example: updating reward weights with outer-loop gradient is not sufficient; periodic resetting reward weights helps to escape from poor local optima.

The limitation of **Gradient** arises from two inherent properties of multi-objective problems: (1) adding competing objectives increases task complexity and expands the search space for reward weights; and (2) task criteria provide extremely sparse feedback, yielding weak gradient signals for backpropagation. Starting with a poor initial weight often leads the policy to converge to a sub-optimal solution, which in turn prevents the outer loop from making meaningful gradient updates. Consequently, gradient-based optimization is only effective when the number of objectives is small, the reward is limited, the solution space is broad, and the mapping from weights to performance is smooth and continuous, avoiding rugged, non-convex landscapes.

However, real-world tasks rarely satisfy these conditions. To illustrate this, we visualize the Walker2d task in the MuJoCo domain in Fig. 2b. Since Walker2d has three objectives, we project the 3D reward weight space into 2D via Principal Component Analysis (PCA), and use the z-axis to represent the optimal task performance achieved under each weight. We randomly sample around 100 weights in the scenario, use these weights to train separate RL policies, and record the final performance of each policy. After interpolating the task performance, we construct the performance surface. The resulting landscape exhibits numerous local optima, violating the assumptions under which **Gradient** succeeds.

Interestingly, our experiments show that randomly resetting reward weights helps escape poor local optima, leading to substantially better performance. We will provide more rigorous empirical validation of these findings in Sec. 6.1.

## 5 MORSE: Multi-Objective Reward Shaping with Exploration

The motivation example suggests that introducing noise into the bi-level optimization can potentially discover superior reward function and improve convergence performance. Inspired by this insight, we present a general framework to enhance bi-level optimization and propose a pratical technique, MORSE. In this section, we present MORSE, a stochastic optimization method that enhances exploration in the outer loop through selective noise injection based on two metrics, reward space novelty and task performance.

### 5.1 Exploration Guided by Task Performance and Reward Space Novelty

We present the outer-loop exploration mechinism in Alg. 1.

Each episode receives a task reward of 1 if it satisfies task success criteria, otherwise 0. We use $P$ to denote task performance, which is the fraction of successful trajectories over all rollouts obtained using the current policy. Exploration is triggered with probability $1 - P$.

Reward-space novelty is measured using a curiosity-driven metric based on Random Network Distillation (Burda et al., 2018). Specifically, we introduce two networks: a fixed, randomly initialized target network $f_{target}$ and a trainable predictor $f_{pred}$, both mapping reward weights to scalar values. Novelty is quantified as the prediction error between them, with larger discrepancies indicating more novel reward weights. The predictor is trained with the reward weights previously used for RL

training, minimizing the MSE loss $= \|f_{\text{pred}}(w(s)) - f_{\text{target}}(w(s))\|_2$. We define the novelty metric for a weight candidate $w$ as $V_{\text{novelty}}(w) = \|f_{\text{pred}}(w(s)) - f_{\text{target}}(w(s))\|_2$. Larger score indicates that $w$ lies in a less explored region.

Outer-loop exploration proceeds by randomly sampling $N$ candidate reward weights $W = \{w_i\}_{i=1}^N$ from the weight space. Each candidate is evaluated by the novelty metric $V_{\text{novelty}}(w_i)$, and a new weight $W'$ is selected via softmax sampling over these scores.

---

**Algorithm 1** Outer Loop Exploration

---

**Require:** History reward weights $W_{past}$, sample size $N$, the current task performance $R_{\text{curr}}$, the variation in task performance between two training epochs $\Delta R$, improvement threshold $\alpha$
1: **if** $\Delta R < \alpha$ **then**
2:     Train the predictor network $f_{pred}$ with $\|f_{\text{pred}}(w) - f_{\text{target}}(w)\|_2, w \in W_{past}$
3:     Let $P_{curr} = \text{normalize}(R_{curr})$. With probablity $1 - P_{curr}$, perform the following steps:
4:     Randomly sample $N$ points $W = \{w_i\}_{i=1}^N$ from the reward weight space
5:     Compute novelty metric: $V_{\text{novelty}}(w_i), \quad \forall w_i \in W$
6:     Sample from W based on softmax probabilities of $V_{\text{novelty}}$: $w \sim \textbf{Softmax}(V_{novelty})$
7: **end if**

---

## 5.2 OVERALL ALGORITHM

Combining the bi-level optimization and outer loop exploration, we present a realization of Multi-Objective Reward Shaping with Exploration, which is summarized in Alg. 2.

In each training epoch, we rollout data under the current policy $\pi_\theta$ and update $\theta$ via any on-policy or off-policy RL optimizer. Every $T_{grad}$ epochs, the multi-objective weight vector $\phi$ is updated by ascending the gradient of $J(\phi)$, and, at a lower explore frequency $T_{explore}$, we reset $\phi$ through outer loop exploration and reinitialize the policy's parameters to encourage the policy to retain high entropy. By jointly optimizing $\theta$ and $\phi$ while periodically reinitializing the policy, MORSE balances efficient exploitation of learned reward compositions with continual exploration in the multi-objective space, yielding robust policies that negotiate conflicting objectives without manual reward engineering.

---

**Algorithm 2** Multi-Objective Reward Shaping with Exploration (MORSE)

---

**Require:** Total training epoch $N_{\text{epoch}}$, RL policy $\pi_\theta$, outer loop update interval $T_{grad}$, exploration interval $T_{explore}$
1: **for** $e = 0$ to $N_{\text{epoch}}$ **do**
2:     **Rollout:** Execute policy $\pi_\theta$ to collect experience
3:     **Update policy:** Perform RL updates to optimize $\pi_\theta$
4:     **if** $e \bmod T_{grad} = 0$ **then**
5:         Update reward weight: $\phi_{e+1} \leftarrow \phi_e + \nabla_\phi J(\phi_e)$
6:         **if** $e \bmod T_{explore} = 0$ **then**
7:             $w_\phi \leftarrow \textbf{Outer\_Loop\_Exploration}(\phi_e)$
8:             **Reset policy:** Reset the whole actor network parameters and the last layer of critic network
9:         **end if**
10:     **end if**
11: **end for**

---

# 6 EXPERIMENTS

## 6.1 PRELIMINARY VALIDATION OF OUTER LOOP EXPLORATION

We validate the effect of outer loop in a simplified environment. Focusing only on the outer loop update and ignoring the inner loop RL training, we treat reward shaping as an optimization problem in a multi-dimensional space. Each reward weight corresponds to a point in the space, and the optimization objective is task performance, represented as a scalar value. For simplicity, we assume

the reward weight is 2D, $x \in [-1, 1]^2$, and task performance is determined by the function $f(x) \in [0, 1]$. The optimization goal is to find the input value that maximizes $f(x)$ within 100 steps.

We design three types of functions with different characteristics to construct varying optimization landscapes: a smooth polynomial function with multiple local optima (SmoothPolynomial), a complex fixed neural network function (FixedNN), and a function with sparse spike regions (RandomSpiky). These functions are normalized to ensure the output is always within the $[0, 1]$ range. For each of the three function types, we randomly generate 10 parameter sets, resulting in a total of 30 functions. We conduct 10 independent optimization runs for each function, starting from different random initial points.

### 6.1.1 TIMING OF EXPLORATION

We evaluate several strategies for resetting the reward weight. **No Exploration** updates the reward weights only by the gradient of the outer loop with no exploration. **Periodic Exploration** samples new reward weights based on novelty scores and resets the reward weights after every 10 gradient updates. **MORSE** reset the reward weight with the novelty metric and performance criteria after every 10 gradient updates.

|  | RandomSpiky | SmoothPolynomial | FixedNN | Average |
|---|---|---|---|---|
| No Exploration | 0.200 (0.422) | **0.832** (0.213) | 0.945 (0.141) | 0.659 (0.432) |
| Periodic Exploration | 0.102 (0.316) | 0.690 (0.214) | 0.695 (0.306) | 0.496 (0.393) |
| MORSE | **0.873** (0.319) | 0.799 (0.273) | **0.946** (0.140) | **0.873** (0.254) |

Table 1: Average performance of different exploration timing across 10 seeds. Each cell reports the mean performance, with standard deviation shown in parentheses. **MORSE** outperforms other baselines in two of the three optimization landscapes.

As shown in Tab. 1, the results show that **MORSE** significantly outperforms all baseline methods. **No Exploration** converges to suboptimal solutions on complex optimization landscapes, such as RandomSpiky. Meanwhile, **Periodic Exploration** reset the reward weights periodically, disrupting the gradient-based optimization process, which leads to worse performance compared to the simple gradient-based method, **No Exploration**. In contrast, **MORSE** can terminate random exploration based on performance criteria, achieving a balance between exploration and gradient optimization.

### 6.1.2 SAMPLING ALGORITHM FOR EXPLORATION

In this set of experiments, we replace RND-based sampling with alternative strategies while keeping all other settings fixed. The baselines include random sampling (random) and two canonical sample-based optimization methods, the cross-entropy method (CEM) (Rubinstein, 1999) and covariance matrix adaptation (CMA) (Hansen et al., 2003). For CEM and CMA, each batch samples five candidate weight vectors, and the top 40% (two elites) are retained to guide the next round.

|  | RandomSpiky | SmoothPolynomial | FixedNN | Average |
|---|---|---|---|---|
| random | 0.701 (0.483) | **0.872** (0.204) | 0.946 (0.140) | 0.840 (0.320) |
| CEM | 0.302 (0.482) | 0.711 (0.273) | 0.770 (0.186) | 0.594 (0.388) |
| CMA | 0.801 (0.422) | 0.698 (0.329) | 0.932 (0.185) | 0.810 (0.330) |
| RND(ours) | **0.873** (0.319) | 0.799 (0.273) | **0.946** (0.140) | **0.873** (0.254) |

Table 2: Average performance of different sampling strategies across 10 seeds. Each cell reports the mean performance, with standard deviation shown in parentheses. RND-based exploration outperforms other baselines in two of the three optimization landscapes.

We present the results in Tab. 2. Compared with random, RND discovers more novel reward weights within a limited number of exploration steps. The performance of CEM and CMA degrades significantly when the number of samples is restricted, as in our setting, or when the optimization landscape is highly non-smooth, as in RandomSpiky. Both methods generate multiple candidates per round and update their sampling strategies based on batch performance. In contrast, our approach samples weights sequentially, one at a time, which substantially reduces the total number of samples required to reach a feasible solution. Since each sample corresponds to a full RL training process, this results in markedly higher computational efficiency.

## 6.2 MORSE IN RL SCENARIOS

We evaluate MORSE in 8 robotic tasks: 3 MuJoCo locomotion tasks, *Halfcheetah*, *Hopper*, and *Walker2d*, and 5 Isaac Sim tasks, including 3 locomotion tasks, *Quadcopter* (Isaac-Quadcopter-Direct-v0), *Unitree-A1-2obj* (modified Isaac-Velocity-Flat-Unitree-A1-v0), *Unitree-A1-9obj* (original Isaac-Velocity-Flat-Unitree-A1-v0), and 2 manipulation tasks, *Reach* (Isaac-Reach-Franka-v0), *Lift* (Isaac-Lift-Cube-Franka-IK-Rel-v0). These tasks involve 2, 3, 3, 3, 2, 9, 4, and 5 reward objectives, respectively.

### 6.2.1 EXPERIMENT SETUP

**Task Criteria and Heuristic Functions.** An episode is considered successful only if the agent survives to the end, reaches the target goal (e.g., specific velocity or position), and its entire trajectory remains within a predefined action threshold. In our experiments, the action threshold is determined by training a policy with oracle weights defined in the codebase and measuring the maximum action features (e.g., torque or angular velocity) of the policy rollouts. For practical applications, these action thresholds are often set empirically, as in (Chen et al., 2025). Heuristic rewards correspond to the single-objective reward components defined in the original codebase. Detailed definitions of task criteria and heuristic functions for each environment are provided in the Appendix.

**Implementation Details.** Given the distinct API and rollout logic between MuJoCo and Isaac Sim, we implement MORSE using two different frameworks, Stable-Baselines3(Raffin et al., 2021) for MuJoCo and rsl-rl(Rudin et al., 2022) for Isaac Sim. In Stable-Baselines3, PPO's critic is extended to a multi-head critic that predicts returns for each heuristic function. In rsl-rl, we keep a single-head critic but store all reward components in the buffer.

**Reward Weight Constraints.** For MuJoCo tasks, we design two difficulty levels. **Easy** setting intialize reward weights randomly within $[0, 1]$ and maintains this range for weight update. In **Hard**, the range of reward weight is $[-1, 1]$. The larger range makes it significantly more challenging to identify the optimal reward function. For Isaac Sim tasks, weights are initialized within $[0, 1]$ and clipped accordingly.

All experiments are repeated 8 times with different random seeds. In the plots, solid lines in the plots denote the means, and shaded regions denote the standard deviations. For more details on hyperparameter settings, please refer to the Appendix.

### 6.2.2 MAIN RESULTS

We compare the following methods. **Oracle**: The policy is trained with the constant reward weight tuned by human expert. **Gradient**: The reward weight is updated using only the bi-level optimization gradient, equivalent to Barfi(Gupta et al., 2023). **MORSE**: The reward weights are reset with MORSE.

Fig. 3 reports the main experiment results. Across all environments, **MORSE** consistently outperforms **Gradient** and achieves performance comparable to **Oracle**, successfully identifying reward weights that enable high policy success rates. Furthermore, the results on MuJoCo-easy and MuJoCo-hard tasks show that, as the reward space expands, the performance of **Gradient** degrades sharply, whereas **MORSE** maintains stable performance. This again validates the insights in Sec. 4: gradient-based optimization method is strongly sensitive to the initial weight selection and the size of exploration space, and is also prone to suboptimal convergence, whereas exploration helps it escape local optima. These findings highlight the critical role of outer-loop exploration in complex robotic tasks and demonstrate the potential of **MORSE** as a general framework for reward shaping.

### 6.2.3 ABLATION STUDIES

MORSE incorporates three key design choices: (1) during gradient-based bi-level optimization, if the policy converges to unsatisfactory performance, the algorithm jumps to a new reward weight; (2) during exploration, the new starting weight is sampled based on the RND novelty metric; (3) when resetting the reward weight, the actor of the RL policy is also reset to encourage sufficient policy exploration.

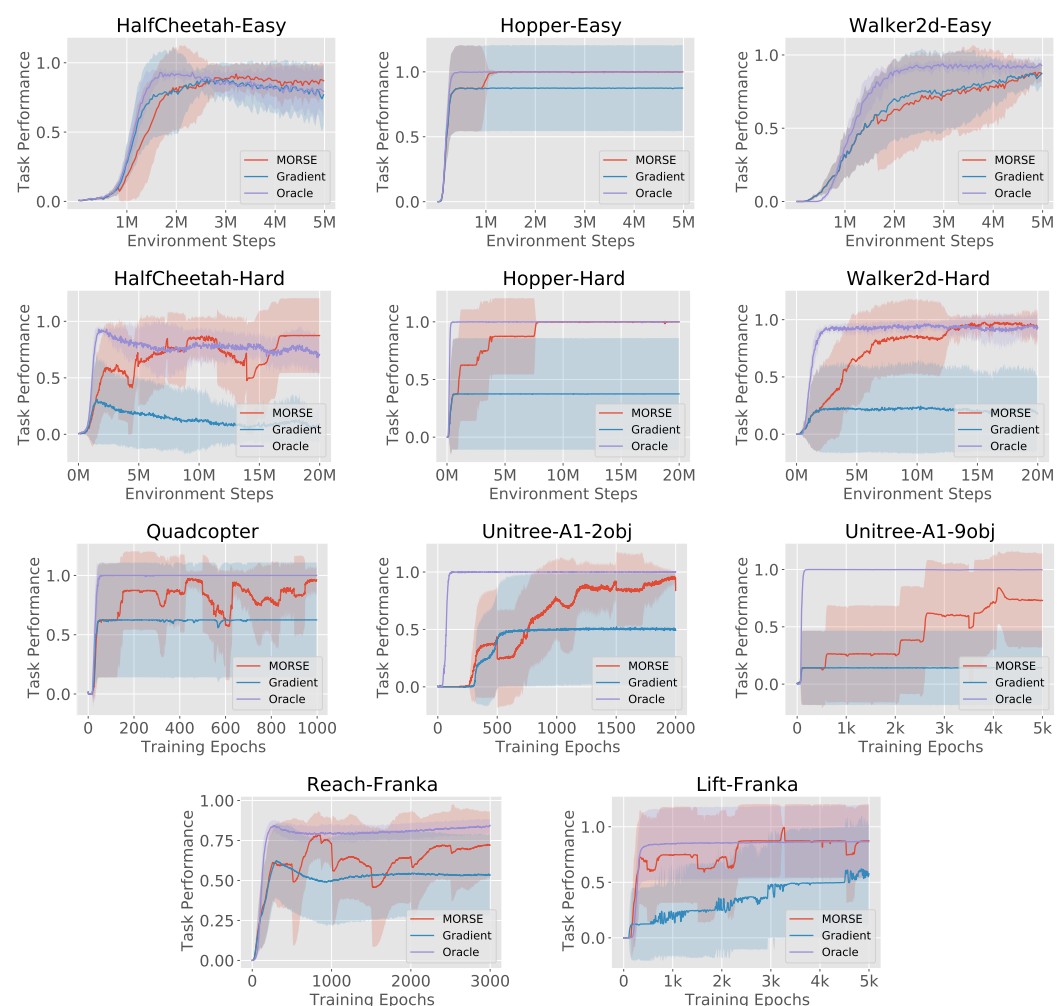

Figure 3: Main results in the MuJoCo and Isaac Sim environments. The results show that **MORSE** outperforms the vanilla bi-level optimization method and is comparable with **Oracle**, whose reward weights are tuned by human expert.

We conduct an ablation study in the MuJoCo-Hard domain, comparing **MORSE** with four variants: **wo/ Gradient**, which conducts outer loop exploration but does not perform gradient update; **wo/ Performance Metric**, which resets weights at fixed intervals, regardless of policy performance; **wo/ Novelty Metric**, which replaces RND with random sampling during exploration; and **wo/ Reset Policy**, which keeps the inner loop policy unchanged when resetting reward weights. Results are shown in Fig. 4.

**wo/ Gradient** performs worse than **MORSE**, suggesting that outer-loop gradient updates help with weight selection. However, **Gradient Only** suggests that, gradient updates are insuffucient and must be complemented with outer loop exploration mechanism.

The discrepency between **MORSE** and **wo/ Performance Metric** highlights the importance of reset conditioning based on task performance. In **wo/ Performance Metric**, the reward weight switches periodically even after it reaches the global optimum, which degrades final performance.

**wo/ Novelty Metric** randomly samples a new reward weight during exploration. For simpler tasks (Hopper) where many reward weights yield high performance, the novelty metric has less impact. However, for harder tasks like Walker2D, selecting the most novel reward weight speeds up the coverage of the whole reward space.

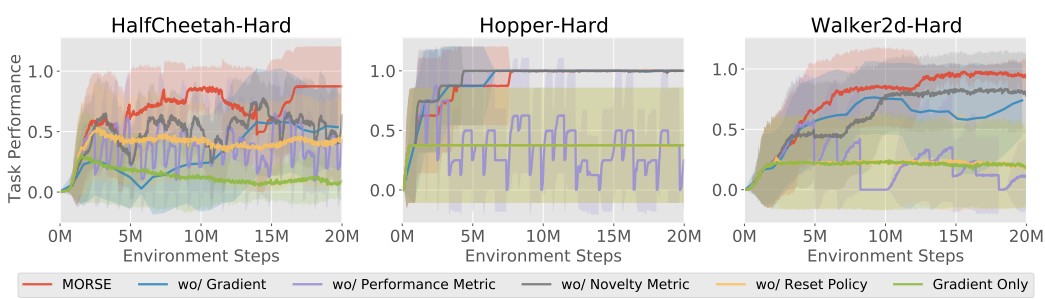

Figure 4: Ablate key components of MORSE in MuJoCo hard domain.

Comparing **wo/ Reset Policy** with **MORSE**, we find that resetting the policy during outer loop exploration is essential for the policy to maintain high entropy and therefore adapt to reward changes.

## 7    CONCLUSION AND FUTURE WORK

Our goal is to reduce the burden of manual reward shaping in RL, simplifying both task design and training. In MORSE, human experts only need to specify sparse task criterion and dense heuristic functions without defining their interactions, while the policy automatically learns to solve complex robotic tasks. We hope this framework can make RL training more scalable and broadly applicable.

There remain several directions for improvement. In outer-loop exploration, novelty and performance metrics could be combined more effectively, for example by considering the historical performance of reward weights and adopting Upper Confidence Bound(Auer et al., 2002) to better balance exploration and exploitation. Moreover, the current design requires resetting the whole policy when switching reward weights. Incorporating successor features(Barreto et al., 2017) into the policy structure may enable more efficient reuse of past knowledge and faster adaptation to new reward weights.

## 8    REPRODUCIBILITY STATEMENT

We have uploaded the source codes in the supplemental materials to ensure reproducibility.

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

## A    PRACTICAL GUIDELINES FOR DEPLOYING MORSE

### A.1    GENERAL SUGGESTIONS

When deploying MORSE, we identify several important conditions which we recommend practitioners to follow.

- Make sure the inner-loop policy converges before performing outer-loop updates (including gradient updates and exploration). For bi-level gradient to hold, the policy must reach its local optimum, otherwise the gradient would be distorted. As for exploration, it relies on the policy's current performance to decide whether to reset weights.

- Since we approximate the Hessian numerically, monitoring its stability is crucial. If the series diverges, the update should be skipped.

- Use slower outer-loop updates. Slowing down outer-loop updates, especially in the early stage of training, improves stability.

- Although MORSE enhances exploration of the reward weight space, extremely high-dimensional settings remain challenging. Stronger priors that narrow the weight range can significantly improve performance. If the search space is large with only a small valid region, more training iterations are required for sufficient exploration.

- Resetting the policy ensures high action entropy under new reward weights, enabling effective exploration.

### A.2    APPLYING MORSE TO REAL WORLD ROBOTIC TRAINING

Domain randomization is widely used to improve sim-to-real transfer of RL policies, but the increased randomness it brings often makes training difficult. Here, we show how to integrate MORSE into a practical training workflow for real-world robotic control.

Rather than applying MORSE directly in a highly randomized environment, we adopt a two-stage training pipeline. Stage 1 performs reward shaping under minimal randomization, which allows MORSE to quickly search for appropriate reward weights. Stage 2 then trains the control policy in a fully domain-randomized environment using the reward function obtained from Stage 1.

We evaluate this pipeline on the IsaacLab quadcopter task. In Stage 1, we randomize only the target position and run MORSE 8 times to obtain 8 different weights. This stage repeats the Quadcopter experiment in Sec. 3. In Stage 2, we add mass and initial pose randomization to the environment, and train RL policies from scratch using both the MORSE-derived weights and the hand-crafted oracle reward. Each reward configuration is trained with PPO for 150 iterations over four seeds.

We present the results in Fig. 5b. Under domain randomization, the reward weights obtained by MORSE achieve performance comparable with the oracle reward. This demonstrates that MORSE-derived reward functions generalize effectively to more challenging training conditions and therefore can be applied to practical robotics settings.

## B    EXPERIMENTAL SETTINGS

### B.1    TASK DEFINITION

In Sec. 6.1, we design three classes of synthetic functions to simulate optimization landscapes of varying difficulty: (1) SmoothPolynomial, a continuously differentiable random polynomial; (2) FixedNN, a function generated by a fixed randomly initialized neural network, which is non-smooth but exhibits no abrupt value shifts; (3) RandomSpiky, a smooth function with sparse sharp peaks. All functions are normalized to produce outputs within the range [0,1]. We present visualizations of representative instances from each class in Fig. 6.

We provide the task definition for RL environments in Tab. 3. In MuJoCo environments, the agent gets a task reward of 1 when it satisfies all task criteria, which are determined using the following protocol. First, we train a policy to convergence using the default reward weights provided in the original environment. Then, we perform multiple rollouts and record the minimum, maximum, and

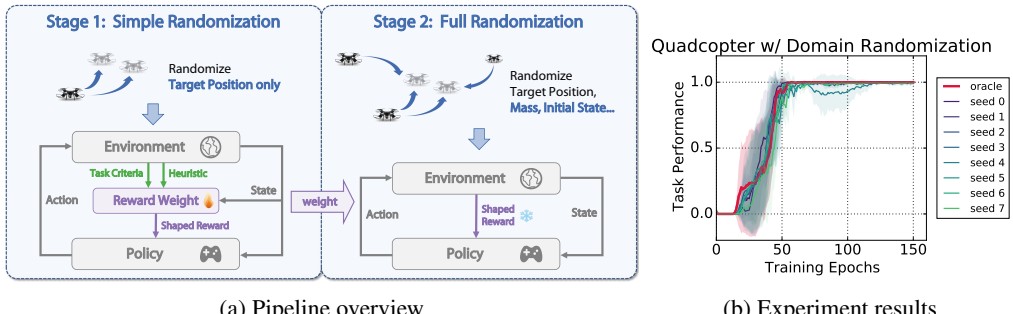

(a) Pipeline overview        (b) Experiment results

Figure 5: (a) 2-stage pipeline for MORSE under domain randomization. (b) Comparison between the oracle reward weight and MORSE-derived reward weights under domain randomization. MORSE reward achieves comparable performance to the oracle reward.

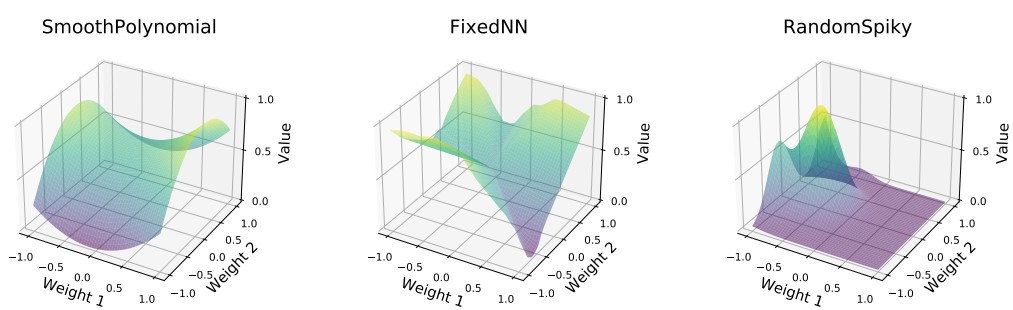

Figure 6: Example optimizaiton landscapes.

mean values of each objective across episodes. Based on these statistics, we define task criteria such that the default policy achieves a success rate exceeding 90%.

To accelerate training, we shorten the episode length and modify the Unitree-A1 task from 9 objectives to 2 objectives. The maximum episode lengths for the environments are 100, 100, 100, 100, 200, 250, respectively.

## B.2 HYPERPARAMETER

We use REINFORCE as the inner-loop optimizer for the CartPole experiments in Sec.4, and PPO for the MuJoCo environments and the Isaac Sim environments in Sec.6.2. Most hyperparameters follow the defaults of their respective implementations. Key hyperparameters are summarized in Table 4.

For CartPole, we use a single environment and set the maximum buffer size to 10,000. The inner-loop REINFORCE actor uses a learning rate of 0.001 and an L2 regularization coefficient of 0.25. The outer loop employs 5-step Neumann approximation with a learning rate of 0.001.

For MuJoCo, we use 32 parallel environments. Each PPO update uses a buffer size of 16,384 environment steps. The actor learning rate is 0.0003, value function coefficient is 0.5, maximum gradient is capped at 1.0, and the entropy coefficient is set to 0.

For Isaac Sim, we use 50 parallel environments. We use adaptive learning schedule for inner-loop policy training. The value function coefficient is 1.0, kl coefficient is 0.01, and the entropy coefficient is set to 0.01. The learning rate for Unitree-A1 is 1e-3, and the learning rate for Quadcopter is 5e-4.

The outer loop applies a softmax-based sampling strategy with temperature $\tau = 10$, where the sampling probability is given by $P(V_{\text{perf}_i}) = \frac{e^{V_{\text{perf}_i} * \tau}}{\sum_j e^{V_{\text{perf}_j} * \tau}}$. To preserve human-designed reward heuristics, we set the learning rate for the reward-weight network to 0.0005 and the coefficient learning

| Environment | Robot | Task Criteria | Heuristic Functions | Weight |
|---|---|---|---|---|
| Cartpole | | Final position $> 0.5$ | Survival reward: 1
Position reward: $\mathbf{1}_{pos>0.5}$
Interference reward: $\mathbf{1}_{action_\pi = action_{PID}}$ | –
–
– |
| HalfCheetah | | Final position $> 5$
control cost $\geq$ -0.6 for each step | Forward reward: $\frac{dx}{dt}$
Control cost: $-\|action\|_2^2$ | 0.4
1 |
| Hopper | | Final position $> 0.8$
control cost $\geq$ -0.004 for each step | Survival reward: 1
Forward reward: $\frac{dx}{dt}$
Control cost: $-\|action\|_2^2$ | 1
1
0.002 |
| Walker2d | | Final position $> 0.8$
control cost $\geq$ -0.006 for each step | Survival reward: 1
Forward reward: $\frac{dx}{dt}$
Control cost: $-\|action\|_2^2$ | 1
1
0.002 |
| Quadcopter | | L2 distance to goal $< 4$
linear velocity $< 100$
angular velocity $< 10000$ | Position reward: $1 - \tanh(\|p - p^*\|^2/0.8)$
Linear velocity cost: $-\Sigma\|v\|^2$
Angular velocity cost: $-\Sigma\|\omega\|^2$ | 15
0.05
0.01 |
| Unitree-A1-2obj | | L2 tracking error $< 0.22$
L2 action rate $< 40$ | Tracking reward: $e^{-(v-v^*)^2}$
Action rate penalty: $-\|a_t - a_{t-1}\|_2^2$ | 1.5
0.01 |
| Unitree-A1-9obj | | L2 tracking error $< 0.22$
L2 action rate $< 40$ | Linear xy velocity error: $e^{-(v_{xy}-v_{xy}^*)^2}$
Angular yaw velocity error: $e^{-(v_{yaw}-v_{yaw}^*)^2}$
Linear z velocity penalty: $-\|v_z\|_2^2$
Angular-$xy$ velocity penalty: $-\|\omega_{xy}\|_2^2$
Joint torque penalty: $-\|\tau\|_2^2$
Joint acceleration penalty: $-\|\ddot{q}\|_2^2$
Action rate penalty: $-\|a_t - a_{t-1}\|_2^2$
Air-time reward: $\Sigma_i(A_i - \tau)F_i,$ if $\|u_{cmd}\| > 0.1$
Non-flat orientation penalty: $-\|g_{b,xy}\|_2^2$ | 1.5
0.75
-2
-0.05
-0.0002
-2.5e-07
-0.01
0.25
-2.5 |
| Reach-Franka | | Position tracking error $< 0.2$
Fine-grained tracking reward $> 0.5$
Orientation tracking error $< 0.6$
L2 action rate $< 100$ | Position tracking error: $\|\mathbf{p} - \mathbf{p}^*\|_2$
Fine-grained pos reward: $1 - \tanh(\|\mathbf{p} - \mathbf{p}^*\|/0.1)$
Orientation tracking error: $d_{quat}(\mathbf{q}, \mathbf{q}^*)$
Action rate penalty: $-\|a_t - a_{t-1}\|_2^2$ | -0.2
0.1
-0.1
-0.0001 |
| Lift-Franka | | L2 object position tracking error $< 0.22$
L2 action rate $< 25000$ | Reaching object reward: $1 - \tanh\left(\frac{\|\mathbf{p}_{obj} - \mathbf{p}_{ee}\|_2}{0.1}\right)$
Lifting object reward: lift $= \mathbf{1}\{z_{obj} > 0.04\}$
Object goal tracking: lift $* \left[1 - \tanh\left(\frac{\|\mathbf{P}^* - \mathbf{P}_{obj}\|_2}{0.3}\right)\right]$
Fine-grained goal tracking: lift $* \left[1 - \tanh\left(\frac{\|\mathbf{P}^* - \mathbf{P}_{obj}\|_2}{0.05}\right)\right]$
Action rate penalty: $-\|a_t - a_{t-1}\|_2^2$ | 1.0
15.0
16.0
5.0
-0.0001 |

Table 3: Task definition for RL environments.

| Environment | Inner Loop
n_epochs | Outer Loop
gradient frequency | reset frequency |
|---|---|---|---|
| Cartpole | 15 | 1 | – |
| HalfCheetah | 10 | 5 | 25 |
| Hopper | 10 | 5 | 25 |
| Walker2d | 20 | 15 | 50 |
| Quadcopter | 5 | 20 | 100 |
| Unitree-A1-2obj | 20 | 20 | 250 |
| Unitree-A1-9obj | 20 | 20 | 250 |
| Reach-Franka | 20 | 20 | 250 |
| Lift-Franka | 20 | 50 | 500 |

Table 4: Hyperparameters for RL environments.

rate to 0.0025. The number of gradient updates for reward weights is adaptive, with a minimum of 3 and a maximum of 10 iterations per outer-loop step, terminating early upon convergence. Similarly, the prediction network is trained using a dynamic number of iterations, from 1 to 1000, to allow it to overfit known data points during each update cycle.

Each run takes less than 1.5 hour on a 4090 GPU.

# C ADDITIONAL EXPERIMENTS AND ANALYSIS

## C.1 ALTERNATIVE DESIGN CHOICES OF MORSE

MORSE is a modular framework, and each component can be flexibly instantiated.

For inner-loop policy training, users can choose any RL algorithms they like, such as SAC(Haarnoja et al., 2018).

For the approximation of outer-loop gradient, while we adopt the implicit function approach following (Gupta et al., 2023), users can apply other meta-learning approaches, such as (Hu et al., 2020). We compare both approaches in MuJoCo-hard environments under the Gradient Only setting as in Sec. 3. As shown in Fig. 7, the two estimators yield similar performance, likely because both are trapped in local optima in the absence of exploration.

MORSE also permits diverse novelty metrics for exploration. Sec. 4 reports results with random sampling, and Fig. 8 further extends the comparison to include Bayesian Optimization (Shahriari et al., 2015). Across these variants, RND achieves the strongest performance, providing more informative exploration signals for reward-weight search.

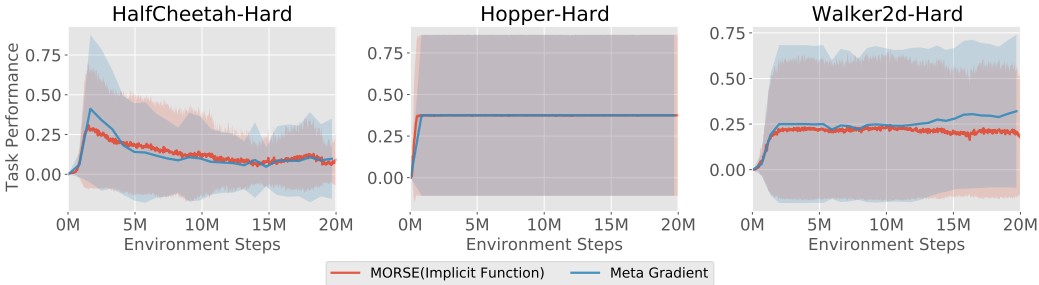

Figure 7: Comparison of implicit-function and meta-gradient estimators for approximating the outer-loop gradient.

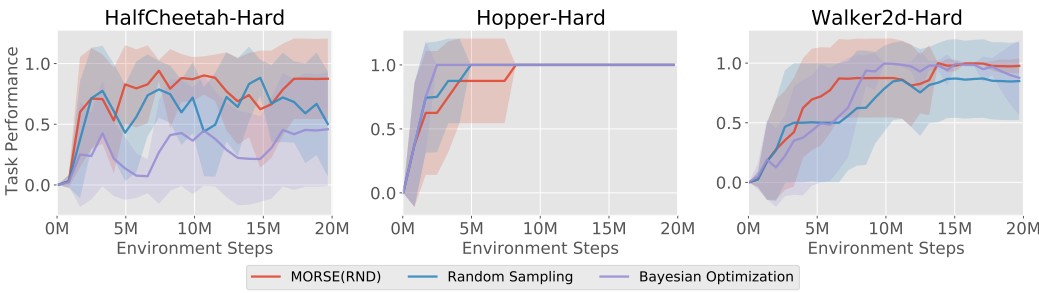

Figure 8: Comparison of RND, random sampling, and Bayesian Optimization as exploration metrics.

## C.2 OPTIMIZATION PATH FOR PRELIMINARY VALIDATION

Following the experiment setting in Sec. 6.1.1, we visualize the optimization path of each experiment in Fig. 9. The results show that **MORSE** can quickly find the most novel reward weight and perform a stable gradient update near the reward weight that performs well, resulting in optimal performance.

## C.3 SENSITIVITY ANALYSIS OF OUTER-LOOP PARAMETERS

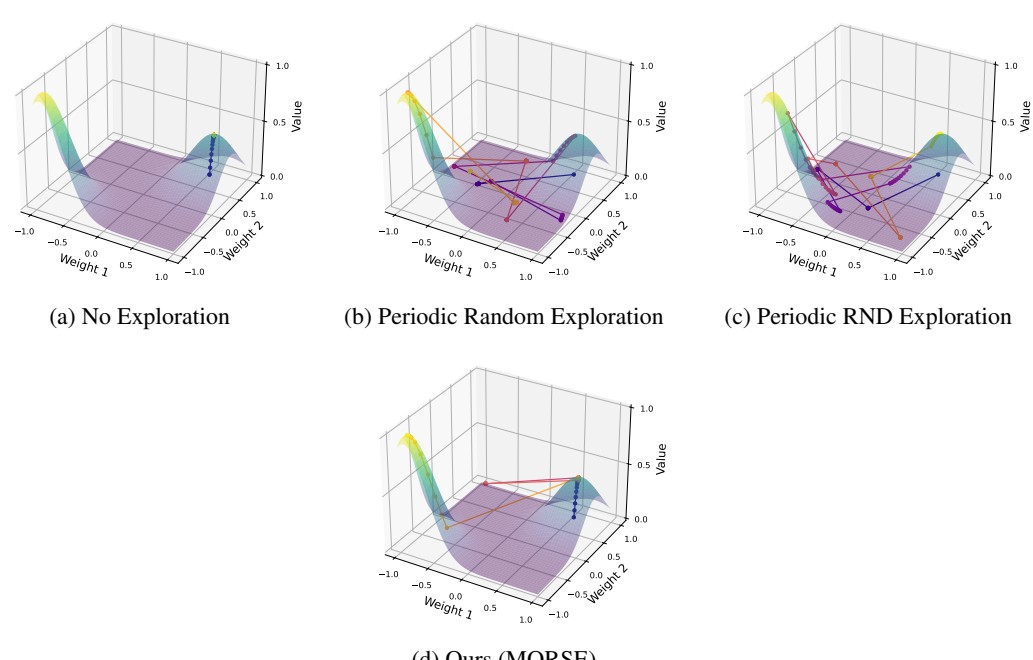

(a) No Exploration      (b) Periodic Random Exploration      (c) Periodic RND Exploration

(d) Ours (MORSE)

Figure 9: Visualization of optimization path. MORSE optimizes towards the global optimum on the lower left.

The outer loop of MORSE has two principal hyperparameters, the bi-level gradient learning rate and the outer-loop exploration interval. Using the same experimental setup as in Sec. 3, we sweep each hyperparameter independently and report the results in Fig. 10 and Fig. 11.

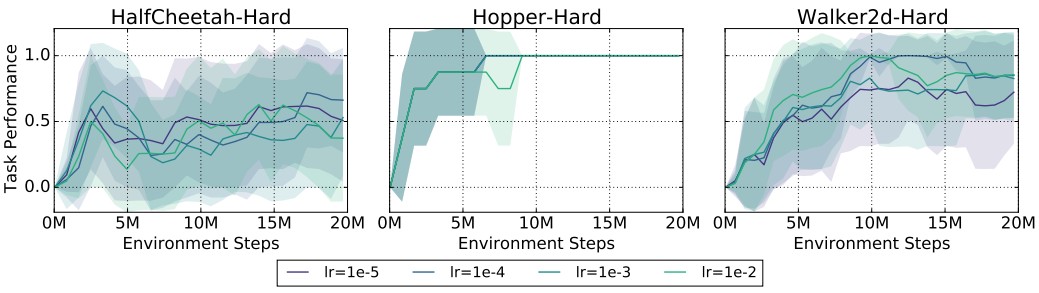

Figure 10: Sensitivity to outer-loop learning rate.

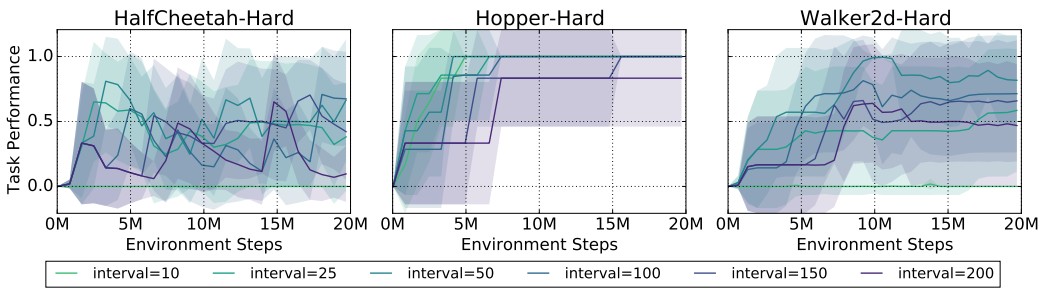

Figure 11: Sensitivity to exploration interval $T_{explore}$.

Our key observations are as follows. First, the overall policy performance is relatively insensitive to the change of learning rate across the tested range, probably due to our constraints on the update process. We clip the reward within its valid range, and use gradient clipping to prevent aggressive updates.

Second, exploration interval can greatly influence performance. When the interval is too small, the outer loop triggers exploration before the inner-loop policy adapts to the current reward weights, causing promising weights to be incorrectly judged as poor solutions and thus discarded. However, when the interval becomes too large, the outer loop can only explore a limited number of times within a fixed training budget, which reduces the chance of discovering high-quality weights. Therefore, we recommend carefully choosing the explore frequency and extending the training budget, so that the inner loop policy has enough time to converge to the current reward before the next exploration, and the outer loop can perform sufficient exploration to find the optimal weight.

### C.4 PCA VISUALIZATION OF WEIGHT SELECTION

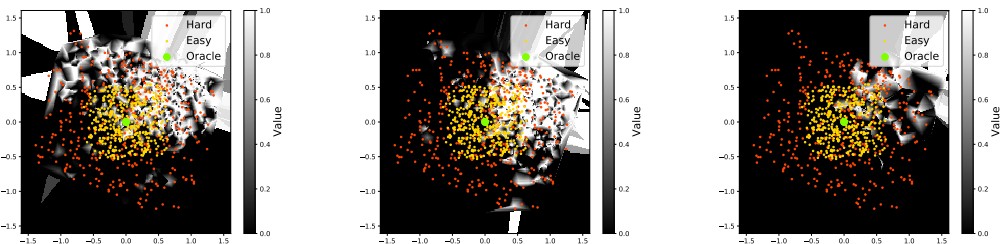

Figure 12: Visualization of the optimization planes of MuJoCo environments using PCA. The color denotes task success rate. The brighter, the higher. From left to right: HalfCheetah, Hopper, Walker2d.

In this section, we visualize the changes of the reward weight with each outer-loop update during the training process in MuJoCo environments. Since the reward weight is three-dimensional, we apply PCA to reduce it to two dimensions for visualization.

In Fig. 12, we illustrate the relationship between reward weight and task success rate across the entire reward weight space. We collect reward weights and success rates from all experiments and then apply interpolation and edge filling to complete the missing regions. We also highlight the human-tuned oracle reward weight and mark the valid reward weight ranges for both easy and hard settings with 400 randomly sampled points. The resulting optimization landscape exhibits multiple local optima, consistent with our hypothesis that robotic task optimization landscapes are often rugged and therefore challenging for gradient-based optimization.

Fig. 13 presents the evolution of reward weight for MuJoCo-Hard environments. We compare four methods: **MORSE**, **Gradient**, and two ablation variants, namely, **wo/ Performance Metric** and **wo/ Novelty Metric**. All methods start optimizing from the same initial point.

**Gradient** is highly sensitive to the choice of initial point. When the initial point falls to undesired areas, gradient updates can be ineffective. **wo/ Performance Metric** escapes the optimal solution. **wo/ Novelty Metric** covers the reward weight space more slowly. For more challenging tasks such as Walker2d, which have lower exploration frequency and fewer reset chances, **wo/ Novelty Metric** limits the ability to find suitable reward weights.

## D THE USE OF LARGE LANGUAGE MODELS (LLMS)

We use LLMs for code debugging and paper polishing.

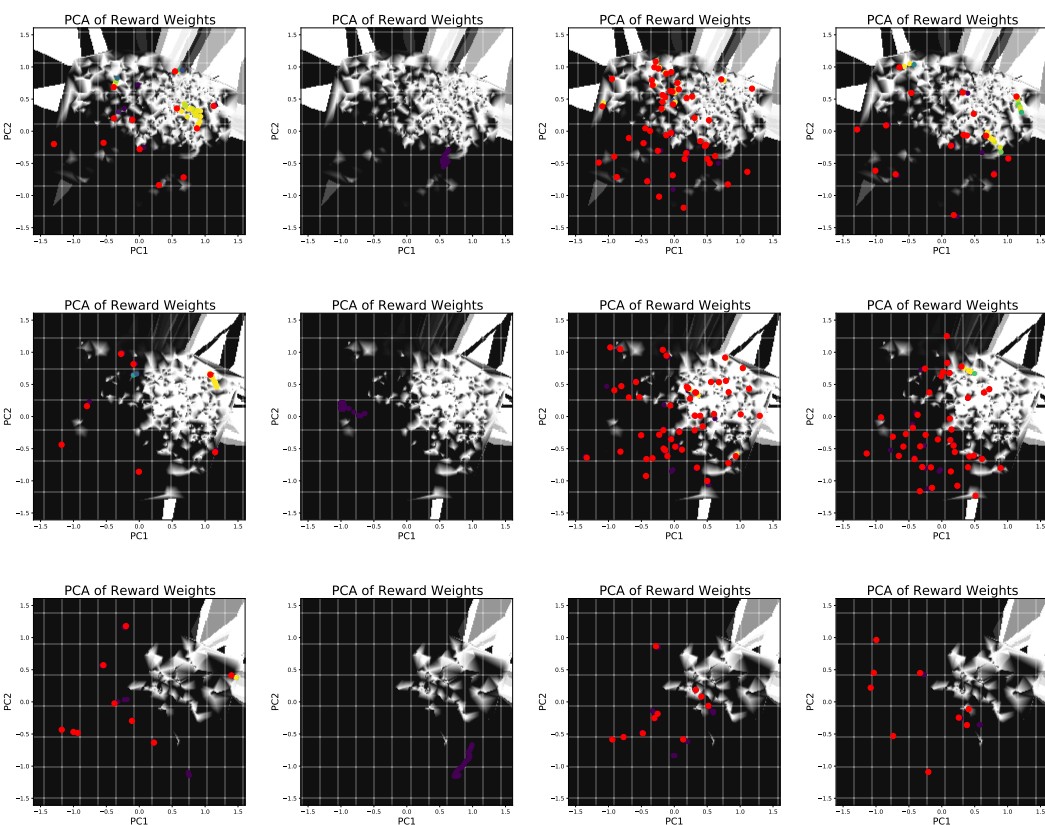

Figure 13: Evolution of reward weights in MuJoCo-Hard environments. Each row corresponds to a different environment, and each column represents a method. From top to bottom: HalfCheetah-Hard, Hopper-Hard, Walker2d-Hard. From left to right: **MORSE**, **Gradient**, **wo/ Performance Metric**, **wo/ Novelty Metric**. The color of the scatter points indicates the number of outer-loop updates, with lighter colors representing more updates. Red points denote new starting points selected when resetting the reward weight.

