# OpenReview forum: "Automatic Reward Shaping from Multi-Objective Human Heuristics"
_ICLR.cc/2026/Conference — Submitted to ICLR 2026_

### Official Review · Reviewer_WoB4 · 2025-10-30

**Soundness:** 2
**Presentation:** 2
**Contribution:** 3
**Rating:** 4
**Confidence:** 4

**Summary:**

The paper proposes MORSE (Multi-Objective Reward Shaping with Exploration), a bi-level optimization framework that automatically combines multiple heuristic rewards into a shaped reward for reinforcement learning. The method introduces stochastic exploration in the outer loop, guided by both task performance and a novelty metric computed via Random Network Distillation (RND). Experiments on several MuJoCo and Isaac Sim locomotion tasks show that MORSE can achieve performance comparable to manually tuned “oracle” rewards while avoiding local minima that affect conventional bi-level optimization.

**Strengths:**

1. The motivation is clear and realistic: reward shaping is indeed a bottleneck in RL, and automating it is a useful and practical direction.

2. The paper is well motivated and the proposed method is technically sound overall, combining bi-level optimization and exploration in a novel way.

3. The approach is close to practice, since many real tasks combine a sparse goal reward with auxiliary shaping terms.

4. The ablation studies are valuable, helping clarify which design elements (exploration trigger, RND novelty, policy reset) matter most.

5. The writing is generally clear, and the figures illustrate the concept well.

**Weaknesses:**

1.  The experiments are restricted to simple locomotion tasks with only 2–3 heuristic components. This is far from realistic use cases with many interdependent objectives. The study would be stronger if it included manipulation or visual tasks (for instance, environments from RLBench, James et al., 2020).

2. While using novelty helps, relying solely on RND is not fully intuitive for reward-space exploration. Methods such as Bayesian Optimization could provide a more principled trade-off between novelty and task performance.

3. Details are missing. Some important definitions are vague. For example, the novelty metric $V_novelty$ is referenced but never formally defined, and the “task criteria” normalization is unclear. This makes it harder to fully reproduce the results. How exactly the P is calculated?

4. The first validation experiment (Sec 6.1.1) is essentially a synthetic numerical optimization example. While it supports intuition, it feels detached from actual RL training and resembles a simulated annealing procedure rather than a meaningful RL benchmark.

5 . Since the method relies on policy resets and outer-loop exploration, it might be sensitive to the learning rate and reset frequency. No robustness analysis is given.

**Questions:**

1.Would off-policy methods (e.g., SAC) benefit even more from this framework, since they can reuse past transitions for multiple reward hypotheses?

2. Could the authors include more diverse environments, such as manipulation or navigation tasks, to test scalability beyond locomotion?

3. How does MORSE compare to a simpler baseline that uses constant reward weights but periodically resets the policy (given that “wo/ Reset Policy” performs poorly in Fig. 4)?

4. How is the oracle reward function defined—are those weights hand-tuned or optimized via grid search?

5. Since learning rate choices affect convergence and local minima, could the authors test MORSE and the baselines under different rates to show robustness?

---

> ### Author Response · Authors · 2025-11-21
>
> # Q1: Experiments in manipulation and high-dimensional tasks
> We add experiments on a 9-objective locomotion task and 2 manipulation tasks. Across these settings, MORSE achieves over 70% success rate under a limited computation budget. Please refer to the meta response or Sec. 6.2.2 Main Results of our updated PDF for detailed information.
> # Q2: Why we use RND instead of Bayesian Optimization; whether SAC can benefit from our framework
> Our main contribution is to show that existing bi-level optimization methods fail for reward shaping and that adding outer-loop exploration significantly improves performance. As stated in Appendix C.1, Alternative Design Choices of MORSE, MORSE is a modular framework, and each of its component can be instantiated differently.
>
> For outer-loop gradient computation, we compare Implicit Function with Meta-Gradient Learning [1] under Gradient Only setting, and find that the two estimators yield similar performance.
>
> For outer-loop exploration strategy, we compare Random Network Distillation (our choice), random sampling, and Bayesian Optimization[2]. We find that RND achieves the strongest performance, providing more informative exploration signals for reward-weight search.
>
> For inner-loop RL training algorithm, while we choose PPO, we believe SAC can be a valid choice, and we are actively integrating SAC into MORSE.  However, SAC requires precise hyperparameter tuning. So far, it can not work reliably in our settings.
> Please refer to the meta response or Appendix C.1 Alternative Design Choices of MORSE in our updated PDF for detailed information.
>
> [1] Hu, Yujing, et al. "Learning to utilize shaping rewards: A new approach of reward shaping." Advances in Neural Information Processing Systems 33 (2020): 15931-15941.
>
> [2] Shahriari, Bobak, et al. "Taking the human out of the loop: A review of Bayesian optimization." Proceedings of the IEEE 104.1 (2015): 148-175.
>
> # Q3: Unclear definitions
> We apologize for the lack of clarity. We have updated Sec. 5.1 to provide formal definitions.
> - $V_{novelty}$ is computed via RND. Given a candidate reward weight $w$, we feed $w$ into a fixed random network and a trainable predictor network. The novelty score is defined as the L2 error between their outputs, $V_{novelty}(w)=||f_{fixed}(w)-f_{predictor}(w)||_2$. A larger prediction error indicates that the candidate weight lies in a region that has not been explored before.
> - Task-criteria normalization is achieved as follows. At the end of each episode, if the task success criteria are satisfied, the episode receives a terminal reward of 1; otherwise, 0. This ensures comparability across reward-weight settings.
> - During evaluation, we rollout $N$ trajectories. $P$ is defined as the proportion of successful trajectories: $P = \frac{\text{ Number of successful trajectories}}{N}$.
>
> # Q4: Why we include a synthetic optimization example (Sec. 6.1.1)
> The goal of Sec. 6.1 is to give a cost-efficient and intuitive analysis of MORSE’s core mechanisms before running expensive RL training. Specifically, Sec. 6.1.1 studies when we should trigger exploration, and Sec. 6.1.2 studies different sampling strategies for weight exploration.
>
> These experiments are not detached from RL experiments. We further validate each observation in RL settings in Sec. 6.2. The correspondence between experiments in Sec 6.1 and Sec 6.2 is as follows. Currently, only CEM/CMA sampling methods miss their RL counterparts, and we plan to add these experiments in follow-up experiments.
>
> | Experiment in Sec 6.1 | Corresponding experiment in Sec 6.2 |
> | :--- | :--- |
> | 6.1.1 No exploration | 6.2.2 Gradient |
> | 6.1.1 Periodic exploration | 6.2.3 w/o Performance Metric |
> | 6.1.2 Random sampling | 6.2.3 w/o Novelty Metric |

---

> > ### Author Response · Authors · 2025-11-21
> >
> > # Q5: Sensitivity to learning rate and reset frequency
> > We conduct sensitivity analysis on these hyperparameters in MuJoCo-Hard settings. We find that MORSE is insensitive to the outer-loop learning rate, whereas exploration frequency has a strong effect and must be chosen carefully. We recommend using a sufficiently large training budget so that (1) the inner-loop policy can converge under the current reward, and (2) the outer loop can explore thoroughly to find good weights. Please refer to the meta response or Appendix C.3 Hyperparameter Sensitivity in our updated PDF for detailed information.
> > # Q6: Ablating the effects of gradient update
> > We add an ablation baseline, w/o Gradient, which removes outer-loop gradient updates and keeps only outer-loop exploration. We find that (1) w/o Gradient performs worse than MORSE, suggesting that outer-loop gradient update is useful; (2) Gradient Only performs worse than MORSE, suggesting that gradient update alone is insufficient and should be complemented with exploration. Please refer to the meta response or Sec 6.2.3 Ablation Studies in our updated PDF for detailed information.
> > # Q7: Definition of oracle reward weights
> > We test on standard RL benchmarks that provide predefined heuristic reward functions and associated weights. We adopt the heuristic functions directly and treat the predefined weights as oracle weights. These weights are often hand-tuned and work well in these environments. The precise definition of heuristic functions and their oracle weights can be found in Table 3 in Appendix B Task Definitions.

---

> ### Comment · Reviewer_WoB4 · 2025-11-26
>
> Firstly, thank you for your detailed response, which has addressed most of my questions. I would like to follow up on a few points:
> 1. In your reply to reviewer gZgz, you mentioned clipping MORSE's reward weights to the [0, 1] range and employing gradient clipping to limit update magnitudes. Did the gradient-based method or other baseline approaches undergo similar clipping?
> 2. In Figure 3, the x-axis labels for the new experiments display the number of training epochs. If all experiments in your setup employed the same number of training epochs, can you confirm that all baseline models and the MORSE model had identical environmental steps?
> 3. How is the reward weights currently intialized and sampled? In log-scale or normal scale?

---

> ### Author Response · Authors · 2025-11-27
>
> Thank you for your feedback! Below are our reply:
>
> 1.
>
> Yes. Except for the Oracle baseline, which uses the predefined weight of the task throughout the training process, all methods use the same weight clipping and gradient clipping technique.
>
> 2.
>
> We apologize for the confusion. The x-axis differs because the two RL libraries we use adopt different logging units. For MuJoCo tasks (Hopper, HalfCheetah, Walker2d), we use Stable-Baselines3, which takes environmental steps as the x-axis, and for IsaacLab tasks (Quadcopter, Unitree-A1, Reach-Franka, Lift-Franka), we use rsl-rl, which takes inner-loop RL training epochs as the x-axis. We follow each library’s convention for clarity and reproducibility.
>
> For different tasks (e.g., Quadcopter & Unitree-A1), the total number of environmental steps differs. For the same task, all methods use exactly the same number of environmental steps and inner-loop RL training epochs. Outer-loop operations  rely entirely on existing rollouts and thus do not induce extra environmental interaction.
>
> 3.
>
> Currently, we sample the weights uniformly within its valid range. The implementation is `torch.rand(num_reward_components) * (high - low) - low`.

---

### Official Review · Reviewer_fjsX · 2025-10-31

**Soundness:** 3
**Presentation:** 3
**Contribution:** 3
**Rating:** 6
**Confidence:** 3

**Summary:**

The authors present MORSE, a framework for reward shaping that automates reward weight tuning. The method seeks to improve exploration of the reward space by guiding the RL policy towards higher task success. Their goal is not to achieve or balance multiple objectives, but rather to achieve a higher task success rate in general.

The authors propose an inner and outer loop algorithm to automatically find reward weights that aim to maximize the task reward. The inner loop trains an RL policy given a reward weight. The outer loop assigns a novel weight to the inner loop to overcome local optima. This is achieved with an exploration-guided formulation using Random Network Distillation.

Overall, I am positive about this work; the idea is simple and well-evaluated. The paper is on the theoretical side, and I would like to see it applied to more practical problems that involve many more reward terms, where the problem of reward tuning becomes truly cumbersome.

**Strengths:**

- The paper is easy to follow and provides sufficient details for reproducibility. The authors also release their code.
- The authors address an important task in RL, finding reward weights for multiple reward terms is a difficult and time-consuming task, and at the same time, it is crucial for successful training.
- The authors run extensive validation on synthetic examples as well as popular benchmarks, and they provide ablation studies for their design decisions.

**Weaknesses:**

- The paper's primary weakness lies in its comparison to existing methods. For me, it is not fully clear what the precise differences and similarities to traditional Multi-Objective Reinforcement Learning (MORL) are. The authors claim to solve a different task, but it is plausible that a standard MORL formulation could serve as a strong baseline with minor modifications. A direct comparison is currently missing. For instance, the "Gradient w/ Reset" baseline is intuitively similar to some MORL training approaches; it would be helpful if the authors extended their discussion on the differences or explained why MORL is not an option for this task.
- Another weakness is the demonstrated scalability of the method. Reward tuning becomes especially cumbersome in high-dimensional problems (e.g., 10s of reward terms), whereas finding weights between 2-3 terms, as done in the paper's evaluations, can often be done relatively quickly. This connects to the question of scalability raised later. Furthermore, this raises the question of how the method would compare to a simple baseline, such as a non-expert’s first guess.

**Minor:**
- Missing references for IsaacSim and MuJoCo

**Questions:**

- Instead of balancing potentially conflicting objectives, the authors assume their task has a clear success measurement. I am still wondering how the formulation would change or differ if the measurement for success were not clearly defined. For example, in the motion imitation literature, MORL formulations have been studied (e.g., Alegre et al., 2025, AMOR: Adaptive Character Control through Multi-Objective Reinforcement Learning). In this task, the measurement for success can be based on different preferences; for instance, sometimes tracking the motion accurately is most important, while other times reducing vibrations is prioritized. The used rewards, in this case, are dense rewards, but the weights between them reflect these different, sometimes subjective, preferences. I am wondering how this view of success, which can vary based on preferences, could be understood in the context of this work.
- Traditional MORL learns optimal policies for any given reward term weight. This would, in theory, allow one to search in the dense auxiliary-reward space for a policy that maximizes the primary-task reward. The authors motivate their method with a modified cartpole example, where we see that setting random weights and following the gradient results in optimal performance. This makes me wonder if a traditional MORL solution would not achieve a similar effect. How do the two approaches truly compare? The MORL formulation seems to provide more flexibility (e.g., allowing a change in the task reward after training and still finding the optimal weights), whereas this method appears more rigid, providing only a task-specific solution.
- Following a similar line of thought, but from another direction: the reward space might differ for different environments (e.g., if domain randomization is applied to mass properties). Since this method ultimately provides a somewhat rigid solution, I was wondering how large the sim-to-real gap might become, ultimately, if I want to deploy my policy to the real world, my reward shaping is usually heavily influenced by this target. While I do not expect the authors to compare simulation results with real-world deployments (though that would be highly preferred), I would be interested to understand how domain randomization influences the reported results. Does the solution remain similar, or does it diverge?
- The number of reward objectives in the evaluated examples is relatively low (max 3 rewards). However, tasks like motion imitation learning can involve 10s of reward terms. How does this method scale to a significantly larger number of terms? The authors mention this limitation in the appendix, but can we quantify when the method starts to break?

---

> ### Author Response · Authors · 2025-11-21
>
> # Q1: Experiments in manipulation and high-dimensional tasks
> We add experiments on a 9-objective locomotion task and 2 manipulation tasks. Across these settings, MORSE achieves over 70% success rate under a limited computation budget. Please refer to the meta response or Sec. 6.2.2 Main Results of our updated PDF for detailed information.
>
> # Q2: Comparison to non-expert's first guess
> We add a baseline corresponding to "non-expert's first guess", where we randomly initialize the reward weight and fix the weight during training. In MuJoCo-hard environments, this baseline cannot achieve satisfiable performance. This indicates that tasks with only 2–3 objectives remain non-trivial and require reward-weight tuning to achieve strong performance.
>
> | | Halfcheetah-hard | Hopper-hard | Walker2d-hard |
> | :--- | :--- | :--- | :--- |
> | Constant Weight | 0.070 (0.158) | 0.375 (0.518) | 0.140 (0.337) |
> | MORSE  | 0.875 (0.331) | 1.0 (0.0) | 0.977 (0.062) |
>
> # Q3: preference-based success definition
> Regarding your first concern, "how the formulation would change or differ if the measurement for success were not clearly defined", we note that, MORSE does not require a strict, discrete 0/1 success criterion. It only needs an outer-loop evaluation signal that indicates whether the current reward function produces a satisfactory policy. In tasks like locomotion or manipulation, this is easy to define. We can use "whether the current velocity reaches target velocity", or "whether the object is lifted near to its target position" as 0/1 success criteria. For motion imitation tasks, MORSE can use a soft, continuous success score.
>
> For example, AMOR uses an external discriminator to evaluate how close the policy’s actions are to the demo. In a similar way, we can define the success score as "the fraction of actions that are close to the demo within a trajectory." This gives us a soft success metric between 0 and 1 and therefore can be solved directly using MORSE. We can also incorporate hard constraints into the task. For example, when the action violates joint safety constraints, the success score becomes zero. In this way, MORSE can find a reward function that satisfies safety constraints but still retains high imitation quality.
>
> Regarding your second concern, "how this view of success, which can vary based on preferences, could be understood in the context of this work, " we propose two ways to incorporate preference selection into MORSE.
>
> 1. Run MORSE multiple times under the same success criteria. Due to the randomness in weight initialization and exploration, different runs naturally yield different reward weights, some of which prioritize tracking accuracy, others vibration reduction. Thus, we can have a set of Pareto-like reward weights, with different weights corresponding to different preferences.
>
> 2. Encode preference directly in the success criteria. We can tighten the thresholds on imitation accuracy or vibration stability depending on our preference, and MORSE will discover different reward weights under different preference definitions.

---

> ### Author Response · Authors · 2025-11-21
>
> # Q4: Comparison between MORSE and traditional MORL
> Traditional MORL and MORSE solve fundamentally different problems, even though both involve multiple reward terms. Traditional MORL learns a policy conditioned on reward weights, i.e., $\pi(a|s,w)$. Given any weight vector, MORL can produce the corresponding optimal policy. However, MORL does not learn how to choose or optimize the weights. It assumes that the appropriate weights or preferences are given at test time. Even in AMOR, the weight or preference is either manually specified or chosen by a learned high-level controller that requires explicit reward signals.
>
> In contrast, our work aims to solve the orthogonal problem. We optimize the weight vector $w$, not a universal $\pi$ conditioned on $w$.  Therefore, to use MORL in our reward shaping setting, one would still need to add an external success signal and a mechanism to improve $w$ toward higher success, which is exactly what MORSE provides.
>
> While MORL offers flexibility, this flexibility also brings issues. For example, MORL requires the users to explicitly specify preference, and MORL policies may include extreme or undesirable behaviors to adapt to unrealistic weights. However, real-world robotic applications often need only one single valid controller that respects physical constraints. Practitioners need one robust solution, not a whole family of policies. In such settings, the rigidity of MORSE is not a disadvantage but a design choice.
>
> We believe that many insights discovered in MORL literature could improve inner-loop learning efficiency of MORSE (e.g., by sharing common experience across weights). We will explore more in this direction in future work.
>
> Finally, because MORSE and MORL optimize fundamentally different objectives, a fair comparison between the methods is non-trivial. If the reviewer can further describe how to adapt MORL for reward-shaping (i.e., how MORL should update $w$ using a success signal), we would be happy to include the baseline.
>
> # Q5: MORSE under domain randomization
> Thank you for raising this scenario. To apply MORSE in real-world robotic settings, we propose a two-stage pipeline. In the first stage, we run MORSE in environments with no or minimal domain randomization to obtain satisfying reward weights. Then, in the second stage, using the reward weights from Stage 1, we train policies under full domain randomization using standard RL algorithms until convergence.
>
> We validate this pipeline in IsaacLab quadcopter task. For stage 1, we randomize only the target position, corresponding to the experiments in Sec. 6.2.2 Main Results. In stage 2, we randomize the quadcopter’s mass and initial state, and train separate RL policies from scratch, using the oracle weight and the weights found by MORSE. The result suggests that reward weights obtained by MORSE can achieve performance comparable with the oracle reward. This demonstrates that MORSE-derived reward functions generalize effectively and therefore can be applied to practical robotics settings. Please refer to the meta response or Appendix A.2 Applying MORSE to Real World Robotic Training in our updated PDF for detailed information.
>
> # Q6: When MORSE breaks
> To discuss MORSE's scalability to high-dimensional reward space, we provide an approximate lower-bound.
>
> Let $D$ denotes the overall reward-weight search space, and $D^+$ denotes the subspace of "valid weights", which are either successful weights that yield high success scores, or weights that can become successful weights via gradient updates.
>
> Using random search, the probability of sampling valid weights and thus achieving high success scores is $D^+ / D$. Thus, the expected number of explorations needed is $D / D^+$. If each exploration interval equals $K$ inner-loop RL updates, then MORSE needs $K × (D / D^+)$ training epochs in total. Theoretically, MORSE can solve any problem if its training budget is larger than this lower-bound estimate.
>
> Because MORSE uses RND-based sampling for weight exploration rather than random search, in practice, it performs better than this worst-case estimate. Nonetheless, when $D$ becomes extremely large and $D^+$ extremely small (i.e., the task has many reward terms but very few feasible weight combinations), the search problem becomes exponentially hard for any optimization method.

---

### Official Review · Reviewer_gZgz · 2025-11-01

**Soundness:** 2
**Presentation:** 2
**Contribution:** 1
**Rating:** 2
**Confidence:** 4

**Summary:**

- The authors propose MORSE, a framework to automatically combine a set of human-provided, unweighted heuristic reward functions into a single (shaped) reward function. The system only requires practitioners to specify these heuristics and a sparse, episodic task performance criterion
- MORSE formulates this as a bi-level optimization problem
  - The inner loop trains an RL policy to maximize the expected return from the current shaped reward
  - The outer loop updates the reward weight parameters to maximize the sparse task performance
- The paper's core premise is that standard gradient-based bi-level optimization fails in this domain because the weight-performance landscape is highly non-convex. MORSE addresses this by introducing a guided stochastic exploration mechanism into the outer loop
- The method is evaluated on a set of simple locomotion tasks

**Strengths:**

- The paper addresses a critical bottleneck in RL for robotics: the labor-intensive and error-prone process of manual reward function design
- Framing the problem as a bi-level optimization is a good approach (also used in other works)
- The use of synthetic 2D optimization functions to isolate and test the outer-loop exploration strategy (Sec 6.1) is a good experimental design concept and the motivating example was clear

**Weaknesses:**

- Tables 1 and 2 report means without any confidence intervals or error bars, making it impossible to validate the significance of the results
- The paper is a methodology paper but is only tested on locomotion tasks. This is not a robotics paper. The method's generality is unproven, and it should have been tested on a broader set of environments
- The experiments use tasks with only 2 or 3 heuristics, despite motivating the problem with a 15-heuristic example (Margolis and Agrawal). The method's performance in the high-dimensional settings where it would be most valuable is unknown
- Can the authors provide experimental validation on manipulation as well? Can the authors provide results for more complex tasks with a higher number of heuristics?
- The claim that RND "discovers more novel reward weights" than random sampling does not have much supporting evidence
- The paper's premise that gradient-based methods are unsuitable for this problem is an oversimplification and ignores the empirical success of such methods in non-convex settings
- The related work section (Sec 2.2) fails to cite or compare against other key papers in automatic reward shaping for robotics [1, 2]
- Tables 1 and 2 are mislabeled, using "Total" instead of "Average" for the final column

References
[1] Ma, Y.J., Liang, W., Wang, G., Huang, D.A., Bastani, O., Jayaraman, D., Zhu, Y., Fan, L. and Anandkumar, A., 2023. Eureka: Human-level reward design via coding large language models. arXiv preprint arXiv:2310.12931.
[2] Zhang, C.B.C., Hong, Z.W., Pacchiano, A. and Agrawal, P., 2024. ORSO: Accelerating Reward Design via Online Reward Selection and Policy Optimization. arXiv preprint arXiv:2410.13837.

**Questions:**

- The task formulation (Sec 3.1) assumes a simple linear combination of heuristics. Why was this restrictive form chosen? What about non-linear combinations or interaction terms (eg $R_{h_1} \times R_{h_2}$) that might be required to model complex objective trade-offs?
- The outer loop updates the reward weights, which can drastically change the optimization landscape for the inner-loop policy. If you take significant gradient steps on the reward weights, does the policy training not suffer from severe instability due to the non-stationary reward function? How is this handled?
- The paper states "the resulting landscape exhibits numerous local optima, violating the assumptions under which Gradient succeeds". While vanilla GD might fail, modern gradient-based optimizers are designed for and show enormous empirical success in non-convex settings. Can the authors provide a more rigorous justification for why this specific problem is intractable for standard gradient-based optimization? Have the authors tried other gradient-based optimization techniques?

---

> ### Author Response · Authors · 2025-11-21
>
> # Q1: Experiments in manipulation and high-dimensional tasks
> We add experiments on a 9-objective locomotion task and 2 manipulation tasks. Across these settings, MORSE achieves over 70% success rate under a limited computation budget. Please refer to the meta response or Sec. 6.2.2 Main Results of our updated PDF for detailed information.
> # Q2: Evidence for “RND discovers more novel reward weights”
> We apologize for the lack of clarity. This statement can be divided into two parts:
> 1. with the same number of samples, RND finds reward weights that lead to more effective RL training;
> 2. RND finds better weights because it explores the weight space more thoroughly.
>
> Evidence for (1):
> - In Sec. 6.1.2, RND outperforms random sampling in simplified landscapes.
> | | RandomSpiky | SmoothPolynomial | FixedNN | Average |
> | :--- | :--- | :--- | :--- | :--- |
> | RND | 0.873 (0.319) | 0.799 (0.273) | 0.946 (0.140) | 0.873 (0.254) |
> | Random samping | 0.701 (0.483) | 0.872 (0.204) | 0.946 (0.140) | 0.840 (0.320) |
>
> - In Sec. 6.2.3 RND (MORSE) significantly improves performance compared to random sampling (w/o Novelty Metric) in RL settings.
> | | Halfcheetah-hard | Hopper-hard | Walker2d-hard |
> | :--- | :--- | :--- | :--- |
> | RND (MORSE) | 0.874 (0.33) | 0.999 (0.003) | 0.945 (0.134) |
> | Random sampling (w/o Novelty Metric) | 0.404 (0.426) | 1.0 (0.0) | 0.811 (0.333) |
>
> Evidence for (2):
> - In Appendix C.2 Fig. 9, we visualize the optimization path of both methods. Comparing panels (b) and (c), we can see that, under the same number of exploration steps, RND covers a larger region of the weight space than random sampling.
>
> Therefore, we believe that RND’s advantage stems from its ability to explore a broader portion of the reward-weight space, and this increases the possibility of identifying effective weights.
>
> # Q3: Missing references in related work
> Thank you for pointing this out. We have added the relevant papers to Sec. 2.2. However, the task formulation in these works differs from ours. They use LLMs to jointly generate heuristic functions and reward weights, whereas our setting assumes fixed heuristic functions and focuses solely on optimizing their weights. Moreover, because reward tuning requires many iterative evaluations, LLM-based reward design incurs significantly higher computation cost, whereas gradient-based and our proposed methods offer more efficient shaping procedures.
> # Q4: Why we restrict to linear combinations of heuristics
> Linear combination of heuristics is the most common reward pattern in RL, widely adopted in major benchmarks such as MuJoCo and IsaacLab, as well as in practical robot training pipelines[1, 2, 3]. Most Multi-Objective RL and reward-shaping literature also adopt this assumption [4, 5].
>
> However, we agree that non-linear combinations are more expressive and may yield better results. We will explore more on extending MORSE to non-linear reward parameterizations in future work.
>
> [1] Margolis, Gabriel B., and Pulkit Agrawal. "Walk these ways: Tuning robot control for generalization with multiplicity of behavior." Conference on Robot Learning. PMLR, 2023.
>
> [2] Ji, Shilong, et al. "JuggleRL: Mastering Ball Juggling with a Quadrotor via Deep Reinforcement Learning." arXiv preprint arXiv:2509.24892 (2025).
>
> [3] Su, Zhi, et al. "Hitter: A humanoid table tennis robot via hierarchical planning and learning." arXiv preprint arXiv:2508.21043 (2025).
>
> [4] Felten, Florian, El-Ghazali Talbi, and Grégoire Danoy. "Multi-objective reinforcement learning based on decomposition: A taxonomy and framework." Journal of Artificial Intelligence Research 79 (2024): 679-723.
>
> [5] Gupta, Dhawal, et al. "Behavior alignment via reward function optimization." Advances in Neural Information Processing Systems 36 (2023): 52759-52791.

---

> ### Author Response · Authors · 2025-11-21
>
> # Q5: Stability under non-stationary reward updates
> To address the instability caused by changing reward weights, MORSE employs two complementary mechanisms, corresponding to the two outer-loop operations.
>
> (1) When the outer loop performs gradient updates, we prevent abrupt reward changes and thus preserve inner-loop stability. Specifically, we clip the reward weights (e.g., to [0, 1]) and apply gradient clipping to limit update magnitude. This ensures the reward function evolves smoothly, and thus keeps policy training stable.
>
> (2) When the outer loop performs exploration (i.e., sampling new weights), the reward function changes substantially, which induces large variations in the value function. This indeed introduces new challenges, but the main problem is not reward non-stationarity; rather, it is the collapse of policy exploration. At this stage, the inner-loop policy has converged to a local optimum and almost lost its ability to explore. Even if the reward function is changed, the value function barely updates, and the policy cannot escape its previous behavioral pattern.
>
> To address this, we reset the actor and the last layer of the critic. This allows the inner loop policy to (1) relearn the value function under the new reward and (2) restore the actor's exploration capability. Thus, the policy can adapt to the new reward landscape. Our experiments confirm that this procedure is both necessary and effective. In Figure 4 in Sec 6.2.3 Ablation Studies, the gray curve represents the ablation without this reset, and it shows a clear performance drop.
>
> # Q6: Why gradient-based methods fail in our setting
> We believe gradient-based bi-level optimization struggles in our settings because our task focuses on reward shaping, whereas prior works focus on reward alignment. This leads to the following differences:
> - Prior works usually start optimization from a relatively good reward weight with only slight misalignment, while in our setting the initial weights are completely random.
> - Prior works often use dense task reward to provide step-level feedback for the outer-loop gradient update. However, our tasks have extremely sparse signals, giving only one task reward at the end of each episode. Furthermore, when the weights are poorly initialized, this signal can be zero, providing no useful guidance at all.
> - Prior works usually optimize 2–3 objectives, whereas our tasks involve up to nine objectives, which increases the difficulty of reward shaping.
>
> To verify this, we add a set of experiments on MuJoCo environments, using oracle reward weights as the initial weights. Gradient perform slightly better than Oracle in this case. This suggests that gradient-based optimization works if it can start from a good initial weight.
>
> | | Halfcheetah-easy | Hopper-easy | Walker2d-easy |
> | :--- | :--- | :--- | :--- |
> | Oracle | 99.609 (1.033) | 83.203 (32.511) | 78.125 (37.206) |
> | MORSE | 90.625 (23.006) | 70.156 (43.688) | 90.0 (29.007) |
> | Gradient | 92.656 (12.666) | 84.844 (30.639) | 95.156 (20.408) |
>
> We evaluate another gradient-based optimization technique, i.e., meta-gradient learning[1], in our setting. This method treats the updated policy $\theta'$ as a differentiable function of reward weight $\phi$, and applies the chain rule to compute how inner-loop policy gradients interact with the outer-loop reward weights. On MuJoCo-hard tasks, the two estimators yield similar performance, likely because both gradient-based methods are trapped in local optima in the absence of exploration. Please refer to the meta response or Appendix C.1 Alternative Design Choices of MORSE in our updated PDF for detailed information.
>
> | | Halfcheetah | Hopper | Walker2d |
> | :--- | :--- | :--- | :--- |
> | MORSE (Implicit Function) | 0.091 (0.159) | 0.375 (0.484) | 0.176 (0.307) |
> | Meta-Gradient Learning | 0.098 (0.253) | 0.375 (0.484) | 0.32 (0.419) |
>
> [1] Hu, Yujing, et al. "Learning to utilize shaping rewards: A new approach of reward shaping." Advances in Neural Information Processing Systems 33 (2020): 15931-15941.

---

### Author Response · Authors · 2025-11-21
**New experiment results and major revisions (Part 1)**

Thank you for your insightful suggestions. We have conducted additional experiments and updated the detailed analysis in the revised PDF. Below we summarize the final results corresponding to the reviewers’ requests. Each cell reports the mean performance, with standard deviation shown in parentheses.
# Main Results (Sec 6.2.2)
We conduct experiments on two IsaacLab manipulation environments, Reach-Franka (*Isaac-Reach-Franka-v0*) and Lift-Franka (*Isaac-Lift-Cube-Franka-IK-Rel-v0*), which contain 4 and 5 objectives. We also evaluate Unitree-A1-9obj (*Isaac-Velocity-Flat-Unitree-A1-v0*), which has 9 objectives. We train Reach-Franka, Lift-Franka, and Unitree-A1-9obj for 1500, 5000, and 5000 iterations, respectively, and report success rates over 8 seeds (mean and std).

Across all tasks, MORSE significantly improves success rates compared to Gradient. We hope these experiments demonstrate the scalability and generality of our method.

| | Reach-Franka |  Lift-Franka | Unitree-A1-9obj  |
| :--- | :--- | :--- | :--- |
| Oracle | 0.844 (0.041) | 0.859 (0.325) | 0.998 (0.001) |
| MORSE | 0.721 (0.209) | 0.871 (0.329) | 0.729 (0.41) |
| Gradient | 0.533 (0.258) | 0.471 (0.472) | 0.143 (0.323) |

# Ablation Studies (Sec. 6.2.3)
We add a **w/o Gradient** variant that removes outer-loop gradient updates and keeps only outer-loop exploration.

**w/o Gradient** performs worse than **MORSE**, suggesting that outer-loop gradient update is useful. **Gradient Only** performs worse than **MORSE**, suggesting that gradient update alone is insufficient and should be complemented with exploration.
| | Halfcheetah-hard | Hopper-hard | Walker2d-hard |
| :--- | :--- | :--- | :--- |
| **MORSE** | 0.874 (0.33) | 0.999 (0.003) | 0.945 (0.134) |
| **w/o Gradient** | 0.537 (0.344) | 1.0 (0.0) | 0.74 (0.304) |
| **Gradient Only** | 0.091 (0.159) | 0.375 (0.484) | 0.176 (0.307) |
# Applying MORSE to Real-World Robotic Training (Appendix, Sec. A.2)
Training with domain randomization can bridge the sim-to-real gap of RL policies but substantially increases training difficulty. Therefore, we propose a 2-stage pipeline to extend MORSE to support domain randomization.

## Pipeline Overview
Our proposed pipeline consists of two stages. In stage 1, we run MORSE in environments with no or minimal domain randomization to obtain satisfying reward weights. Then, in stage 2, using the reward weights from Stage 1, we train policies under full domain randomization with standard RL algorithms to convergence.

## Experiments on IsaacLab Quadcopter
For stage 1, we randomize only the target position and run MORSE for 8 seeds, collecting 8 sets of reward weights. This corresponds to the experiments in Sec. 6.2.2 Main Results. In stage 2, we randomize the quadcopter’s mass and initial state, and train RL policies from scratch using the oracle weight and the 8 weights found by MORSE. Each weight is tested 4 times for 150 iterations to convergence.

Under domain randomization, the reward weights obtained by MORSE achieve performance comparable with the oracle reward. This demonstrates that MORSE-derived reward functions generalize effectively and therefore can be applied to practical robotics settings.
| Oracle | MORSE 1 | MORSE 2 | MORSE 3 | MORSE 4 | MORSE 5 | MORSE 6 | MORSE 7 | MORSE 8 |
| :--- | :--- | :--- | :--- | :--- | :--- | :--- | :--- | :--- |
| 1.0 (0.0) | 0.998 (0.004) | 0.99 (0.016) | 0.997 (0.004) | 1.0 (0.0) | 0.993 (0.011) | 1.0 (0.0) | 0.995 (0.009) | 0.996 (0.006) |

---

> ### Author Response · Authors · 2025-11-21
> **New experiment results and major revisions (Part 2)**
>
> # Alternative Design Choices of MORSE (Appendix, Sec. C.1)
> MORSE is a modular framework, and each of its component can be instantiated differently.
> ## Outer-loop gradient computation
> For outer-loop gradient approximation, we compare Implicit Function (our choice) with Meta-Gradient Learning [1] under Gradient setting. The two estimators yield similar performance, likely because both are trapped in local optima in the absence of exploration.
> | | Halfcheetah-hard | Hopper-hard | Walker2d-hard |
> | :--- | :--- | :--- | :--- |
> | MORSE (Implicit Function) | 0.091 (0.159) | 0.375 (0.484) | 0.176 (0.307) |
> | Meta-Gradient Learning | 0.098 (0.253) | 0.375 (0.484) | 0.32 (0.419) |
>
> ## Outer-loop exploration strategy
> We compare Random Network Distillation (our choice), random sampling, and Bayesian Optimization[2] as outer-loop exploration strategy.  Across these variants, RND achieves the strongest performance, providing the most informative exploration signals for weight search.
> | | Halfcheetah-hard | Hopper-hard | Walker2d-hard |
> | :--- | :--- | :--- | :--- |
> | MORSE (RND) | 0.875 (0.331) | 1.0 (0.0) | 0.977 (0.062) |
> | Random Sampling | 0.502 (0.433) | 1.0 (0.0) | 0.85 (0.327) |
> | Bayesian Optimization | 0.459 (0.467) | 1.0 (0.0) | 0.877 (0.314) |
>
> [1] Hu, Yujing, et al. "Learning to utilize shaping rewards: A new approach of reward shaping." Advances in Neural Information Processing Systems 33 (2020): 15931-15941.
>
> [2] Shahriari, Bobak, et al. "Taking the human out of the loop: A review of Bayesian optimization." Proceedings of the IEEE 104.1 (2015): 148-175.
> # Hyperparameter Sensitivity (Appendix, Sec. C.3)
> We analyze MORSE’s sensitivity to the outer-loop learning rate and exploration interval in MuJoCo-hard tasks.
> ## Outer-loop learning rate
> MORSE is relatively insensitive to the change of learning rate, probably due to our constraints on the update process.
> | | Halfcheetah-hard | Hopper-hard | Walker2d-hard |
> | :--- | :--- | :--- | :--- |
> | 1e-5 | 0.508 (0.465) | 1.0 (0.0) | 0.723 (0.392) |
> | 1e-4 | 0.662 (0.397) | 1.0 (0.0) | 0.85 (0.323) |
> | 1e-2 | 0.53 (0.444) | 1.0 (0.0) | 0.826 (0.328) |
> | 1e-2 | 0.373 (0.482) | 1.0 (0.0) | 0.855 (0.327) |
> ## Exploration interval
> Exploration interval can greatly influence performance. When the interval is too small, the outer loop changes the weight before the inner-loop policy adapts to it, causing promising weights to be incorrectly judged as poor solutions and thus discarded. However, when the interval becomes too large, the outer loop can only explore a limited number of times within a fixed training budget, which reduces the chance of discovering high-quality weights.
> | | Halfcheetah-hard | Hopper-hard | Walker2d-hard |
> | :--- | :--- | :--- | :--- |
> | 10 | 0.0 (0.0) | 1.0 (0.0) | 0.0 (0.0) |
> | 25 | 0.382 (0.417) | 1.0 (0.0) | 0.585 (0.472) |
> | 50 | 0.664 (0.47) | 1.0 (0.0) | 0.817 (0.343) |
> | 100 | 0.674 (0.362) | 1.0 (0.0) | 0.714 (0.452) |
> | 150 | 0.422 (0.37) | 1.0 (0.0) | 0.659 (0.466) |
> | 200 | 0.096 (0.215) | 0.833 (0.373) | 0.469 (0.473) |
> # Other Major Revisions
> - In Sec. 2 Related Work, we restate our task setting and its distinction from traditional MORL methods, and add reward shaping literature.
> - In Sec. 5.1, we update and clarify notation.
> - In Sec. 6.1, tables now include both mean and standard deviation.

---

### Comment · Area_Chair_b4nb · 2025-11-26

Dear Reviewers,

Thank you for sharing your valuable insights and expertise, which have played an important role in the review process.

In response to the initial feedback, the authors have submitted a detailed rebuttal addressing the comments raised by the reviewers.

I would appreciate it if you could carefully review their response and consider how it may affect your initial evaluation.

Please feel free to share your updated thoughts or any additional comments after reviewing the rebuttal.

Thank you again for your time and contributions.

---

### Meta-Review · Area_Chair_qQVV · 2026-01-11

**Summary:**

- Reviewers noted that the initial experiments were restricted to simple locomotion tasks with only 2–3 heuristic components, which did not sufficiently the propose method's utility for complex tasks.
- The submission was criticized for not comparing against modern gradient-based optimizers or alternative exploration strategies like bayesian optimization (later added for 3 tasks).
- There were concerns regarding how MORSE fundamentally differs from traditional Multi-Objective RL  and whether a standard MORL formulation could serve as a stronger baseline.

**Reviewer Concerns:**

Addressed:
- The authors added experiments on a 9 objective locomotion task and two manipulation tasks (Reach-Franka and Lift-Franka), demonstrating that MORSE can scale up in this domain at least.
- New results were provided comparing MORSE against Meta Gradient Learning and Bayesian Optimization.
- The authors included a robustness analysis regarding outer-loop learning rates and exploration intervals.

Outstanding:
- The justification for why bi-level optimization fails in reward shaping remains largely empirical rather than theoretically grounded.
- While the new experiments were appreciated, they did not lead to a unanimous consensus toward acceptance, as the core methodology was still viewed by some as incremental (e.g. by combining Bi-level optimization + RND).

**Reviewer Scores:**

- gZgz: This reviewer addressed their request for high-dim tasks and citations, but may still find the non-convexity argument simplified.
- fjsX: This reviewer's primary request for more practical/manipulation problems was met with positive results.
- WoB4: The detailed hyper-parameter sensitivity and new environments directly addressed.

---

### Decision · Program_Chairs · 2026-01-26

Reject